# A SMARTTR workflow for multi-ensemble atlas mapping and brain-wide network analysis

**Michelle Jin[1,2], Simon O Ogundare[1,3], Marcos Lanio[1,4], Sophia Sorid[3], Alicia Ruth Whye[3,5], Sofia Leal Santos[6,7], Alessandra Franceschini[7,8], Christine Ann Denny[7,9]***

[1]Medical Scientist Training Program (MSTP), Columbia University Irving Medical Center (CUIMC), New York, United States; [2]Neurobiology and Behavior (NB&B) Graduate Program, Columbia University, New York, United States; [3]Columbia College, New York, United States; [4]Adult Neurology Residency Program, Stony Brook Medicine, Stony Brook, United States; [5]Tri-Institutional MD-PhD Program, Weill Cornell Medicine, New York, United States; [6]Life and Health Sciences Research Institute (ICVS), School of Medicine, University of Minho, Braga, Portugal; [7]Department of Psychiatry, Columbia University Irving Medical Center (CUIMC), New York, United States; [8]European Laboratory for Non-linear Spectroscopy (LENS), University of Florence, Florence, Italy; [9]Division of Systems Neuroscience, Research Foundation for Mental Hygiene, Inc (RFMH)/New York State Psychiatric Institute (NYSPI), New York, United States

**\*For correspondence:**
cad2125@cumc.columbia.edu

## eLife Assessment

This manuscript describes methods and software, called SMARTR, to map neuronal networks using markers of neuronal activity. They illustrate their approach using tissue from mice that have undergone behavioral tasks. The reviewers considered the study **important** to the field and **compelling** in that the methods and analyses were an advance over current tools.

**Abstract** In the last decade, activity-dependent strategies for labeling multiple immediate early gene ensembles in mice have generated unprecedented insight into the mechanisms of memory encoding, storage, and retrieval. However, few strategies exist for brain-wide mapping of multiple ensembles, including their overlapping population, and none incorporate capabilities for downstream network analysis. Here, we introduce a scalable workflow to analyze traditionally coronally sectioned datasets produced by activity-dependent tagging systems. Intrinsic to this pipeline is simple multi-ensemble atlas registration and statistical testing in R (SMARTTR), an R package which wraps mapping capabilities with functions for statistical analysis and network visualization, and support for import of external datasets. We demonstrate the versatility of SMARTTR by mapping the ensembles underlying the acquisition and expression of learned helplessness (LH), a robust stress model. Applying network analysis, we find that exposure to inescapable shock (IS), compared to context training, results in decreased centrality of regions engaged in spatial and contextual processing and higher influence of regions involved in somatosensory and affective processing. During LH expression, the substantia nigra emerges as a highly influential region that shows a functional reversal following IS, indicating a possible regulatory function of motor activity during helplessness. We also report that IS results in a robust decrease in reactivation activity across a number of cortical, hippocampal, and amygdalar regions, indicating suppression of ensemble reactivation

may be a neurobiological signature of LH. These results highlight the emergent insights uniquely garnered by applying our analysis approach to multiple ensemble datasets and demonstrate the strength of our workflow as a hypothesis-generating toolkit.

## Introduction

Immediate early genes (IEGs), such as *Fos* or *Arc*, are a class of genes that rapidly transcribe and express following recent neuronal activity (*Hunt et al., 1987*; *Lyford et al., 1995*). Peak IEG responses range from the scale of minutes for RNA expression to minutes and hours for protein expression (*Guzowski et al., 2005*). Because of its time-dependent nature, IEG expression has often been used as a proxy for neural activity. As such, high-throughput mapping of IEGs has been immensely useful in relating complex behaviors in rodents to their underlying brain activation (i.e., ensemble) patterns. The advent of effective intact tissue clearing methods (*Chung et al., 2013*; *Renier et al., 2014*; *Susaki et al., 2015*; *Ueda et al., 2020*), microscopy advances (*Ragan et al., 2012*; *Reynaud et al., 2015*), and automated registration and cell segmentation pipelines (*Renier et al., 2016*) has made brain-wide activity mapping projects both scalable and feasible. Recent studies have applied these methods to study brain activation during a wide variety of experiences, including after psilocybin, ketamine, and haloperidol administration (*Davoudian et al., 2023*; *Renier et al., 2016*), the incubation of palatable food craving (*Madangopal et al., 2022*), and fear memory (*Franceschini et al., 2023*). However, with a few exceptions, almost all of these approaches have focused on mapping *single* ensembles across the brains.

Over the last decade, activity-dependent tagging approaches, such as the tamoxifen-inducible ArcCreER$^{T2}$ (*Denny et al., 2014*) and TRAP1/TRAP2 systems (*DeNardo et al., 2019*; *Guenthner et al., 2013*), or the doxycycline-controlled TetTag system (*Reijmers et al., 2007*), have allowed for the investigation of multiple ensembles—and their shared overlap—during two distinct experiences within the same subjects. Such tagging methods have generated unprecedented insight into the neural mechanisms of memory formation, storage, and retrieval by allowing for the identification of individual memory traces. A memory trace or 'engram' (*Semon, 1921*) refers to the enduring physical substrate of memory and is defined as a neural population initially active during memory encoding whose reactivation at a later time point results in memory retrieval. However, the majority of engram studies have focused on a few focal regions, such as the dentate gyrus (DG; *Denny et al., 2014*), retrosplenial area (RSP; *Cowansage et al., 2014*), medial prefrontal cortex (*Kitamura et al., 2017*), and amygdalar subdivisions, such as the lateral amygdalar nucleus (LA), and basolateral amygdalar nucleus (BLA; *Reijmers et al., 2007*), rather than the investigation of numerous regions across the brain. In part, this is because high-throughput mapping and analysis of *multiple* ensembles poses many challenges: (1) Both activity markers need to be clearly immunolabeled across all brain structures under investigation; (2) both labeled markers need to be accurately segmented or automatically counted; (3) a validated strategy needs to be employed for identification of overlapping cell populations; (4) all cell counts and co-labeled cells need to be accurately registered to a standardized atlas space; (5) an infrastructure facilitating easy data management, aggregation, and transformation (e.g., normalizing cell counts by volume) needs to exist; and (6) analytical and visualization methods to intuit functional connectivity between brain regions and how they differ between experimental groupings should be easily implemented.

Here, we developed a workflow incorporating these features to make large-scale analysis of traditionally coronally sectioned datasets produced by activity-dependent tagging systems more accessible. Intrinsic to this pipeline is SMARTTR, a versatile R package that wraps mapping capabilities to the common coordinate framework (CCF) of the Allen Brain Institute (*Wang et al., 2020*) with functions for statistical analysis and network visualization for multiple ensembles using graph theory. Moreover, this package is versatile, such that the analysis modules can be easily applied to imported mapped datasets from other alternative imaging, segmentation, and registration workflows, including those reliant on alternative atlas ontologies, such as the Unified Anatomical Atlas (*Chon et al., 2019*). While we have designed and validated this pipeline with the ArcCreER$^{T2}$ tagging system (*Denny et al., 2014*), the steps and analysis software used in this pipeline are generalizable to virtually all widely used dual-ensemble tagging systems.

Beyond its direct application to engram research, this pipeline has broader use in allowing for the comparison of brain-wide activity patterns during any two distinct experiences. To demonstrate the versatility of our workflow, we applied this approach to gain novel insight into brain-wide functional activation patterns underlying the acquisition and expression of learned helplessness (LH), a robust behavioral paradigm which models traumatic stress. To our knowledge, wide-scale mapping of ensembles underlying both of these time points has not been studied in the context of LH and would allow for greater functional insight into changes occurring between these stages. In this effort, we have identified several altered functional connectivity signatures of the ensembles active during exposure to IS, escapable shock (ES), as well as the shared ensemble population between these experiences. Finally, we have presented this work as a reference dataset freely available for download and walkthrough alongside an accompanying tutorial.

## Results

### Pipeline overview

Our pipeline was designed for mapping and analysis of dual-ensemble datasets generated by the ArcCreER^T2 tagging system (*Denny et al., 2014*), although the steps are, in principle, generalizable to all dual-ensemble tagging approaches. The ArcCreER^T2 mouse line leverages the activity-dependent expression of the IEG *Arc* to enable *permanent* fluorescent tagging of neural ensembles with enhanced yellow fluorescent protein (eYFP) (*Figure 1A*; *Denny et al., 2014*). In this system, injection of a selective estrogen receptor agonist, such as 4-hydroxytamoxifen (4-OHT), opens a transient window for labeling recently active neurons with eYFP. Staining for IEG protein expression (i.e., c-Fos) at a second time point allows for the identification of active ensembles driven by two different behavioral experiences (*Figure 1B*). Following tissue perfusion and brain extraction, multiple ensemble mapping through the pipeline workflow was organized into three different components: (1) immunolabeling and imaging acquisition, (2) image preprocessing and automatic segmentation of dual ensembles and their overlap in ImageJ/FIJI, and (3) atlas registration and downstream analyses in our software package in R (*Figure 1C*).

For the immunolabeling and imaging acquisition, we employed an optimized approach for staining Arc-expressing eYFP⁺ cells and c-Fos⁺ cells in thick serial coronal sections based on our previous work (*Leal Santos et al., 2021*; *Pavlova et al., 2018*). A traditional serial sectioning approach was used, as it is the most accessible and reliable way to stain two markers at a high signal-to-noise ratio across deep brain structures. Following imaging for eYFP⁺ (channel 1) and c-Fos⁺ (channel 2) cells using a scanning confocal microscope (Leica TCS SP8, Leica Microsystems Inc, Wetzlar, Germany) (*Figure 1D*), we implemented a custom segmentation strategy in ImageJ/FIJI which uses a suite of open-source image processing plugins (*Ollion et al., 2013*) for accurate cell segmentation in 3D based on the staining morphology characteristics of eYFP⁺ and c-Fos⁺ cells (*Figure 1E*). The characteristics and positions of segmented eYFP⁺ and c-Fos⁺ cells were saved into .txt files as outputs. Next, co-labeled (eYFP⁺/c-Fos⁺) cells were identified by comparing all possible overlapping objects using the open-source 3D MultiColoc plugin (*Ollion et al., 2013*), saving the output as .txt files, and thresholding by percent volume overlap downstream. The segmentation and colocalization algorithms were efficiently applied in-batch using custom-written ImageJ/FIJI macros that are freely available for download (*Source code 3*, *Source code 4*, *Source code 5*). Importantly, these macros have been generalized to allow for easy parameter modification through a graphical user interface (GUI), to account for different imaging resolutions and unique naming conventions.

In the final preprocessing step, since registration is performed on single-plane .tiff images, we collapsed raw images across all z-planes and channels to use as an aggregate reference image for registration to atlas templates from the Allen Institute Mouse Brain Atlas (*Lein et al., 2007*; *Oh et al., 2014*). This step was also batch processed through a generalized macro in ImageJ/FIJI that is available for download.

### The SMARTTR package for registration, analysis, and visualization of multiple ensembles

The SMARTTR package (simple multi-ensemble atlas registration and statistical testing in R) includes a high-level application programming interface (API) which facilitates interactive user-friendly registration,

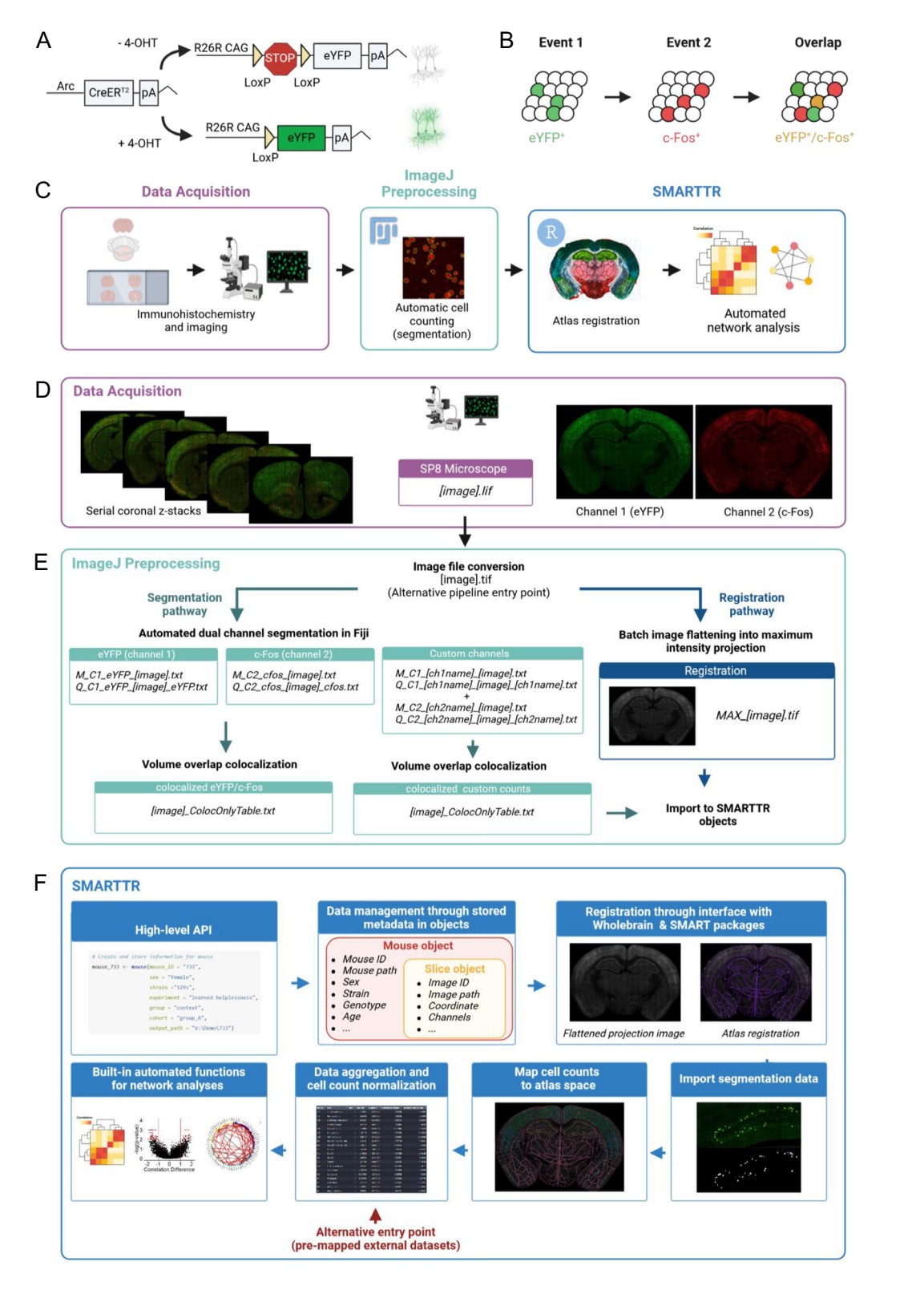

**Figure 1.** Pipeline schematic. (**A**) The ArcCreER[T2] × eYFP tagging strategy allows for labeling of Arc-expressing cells with eYFP following injection of 4-hydroxytamoxifen (4-OHT). (**B**) Indelible labeling of Arc+ ensembles with eYFP followed by immunolabeling for c-Fos+ allows for identification of a co-labeled ensemble active during two distinct time points. (**C**) A graphical summary of the components of the workflow. (**D**) eYFP+ and c-Fos+ cells are immunolabeled and imaged across brain-wide coronal sections. (**E**) eYFP+ and c-Fos+ populations are automatically segmented, and co-labeled

*Figure 1 continued on next page*

*Figure 1 continued*

cells are identified in ImageJ/Fiji. Images are automatically preprocessed and flattened for registration alignment downstream. (**F**) The object-oriented infrastructure of the SMARTTR package in R allows for importation of segmentation data, registration, and mapping, and statistical analysis and visualization using a user-friendly API. This figure was created with BioRender.

mapping of cell counts, and easy statistical analysis and visualization of multiple ensemble datasets (*Figure 1F*). This standard API was designed around an object-oriented infrastructure. In the package, data is managed using slice, mouse, and experiment objects which also store additional metadata in a hierarchical manner. For example, a slice object contains all imaging-related metadata such as the image ID, image path, channel names, and a matching anterior–posterior atlas coordinate, alongside registration data and imported segmentation data. Slice objects are hierarchically stored inside mouse objects, which contain additional metadata such as mouse ID, sex, strain, and genotype. Package functions for interactive registration, importation of segmentation outputs, mapping cell counts to a standard atlas space, automated network analysis, and visualization operate directly on these objects. Moreover, there are also a suite of additional capabilities, including data quality checking for outlier counts, removal of regions to omit due to tissue damage, and normalizing co-labeled counts by a particular channel. The organization of the SMARTTR package also facilitates alternative pipeline entry points. Import functions exist for alternate segmentation approaches. Compatibility with data-sets previously registered and mapped with alternative workflows (*Eastwood et al., 2019*; *Jin et al., 2022*; *Renier et al., 2016*; *Yates et al., 2019*) is supported (*Figure 1F*), allowing users to take advantage of the analysis and visualization capabilities of this package without reliance on prior processing steps. Extensive documentation of the package functions and a detailed tutorial using the datasets from this study are available at the package website (https://mjin1812.github.io/SMARTTR/).

## Identifying ensembles underlying the acquisition and expression of LH

To demonstrate the strengths and capabilities of our dual-ensemble mapping pipeline, we applied our approach to gain novel insights into the neural activity patterns underlying LH, a robust behavioral paradigm widely used to model aspects of trauma and depression preclinically. The neural correlates of LH have been studied across a variety of model organisms (*Maier and Seligman, 2016*) and have been largely focused on single or few regions. More recently, there have been investigations into the brain-wide neural activity patterns following LH expression (*Kim et al., 2016*). However, to our knowledge, the neural ensembles underlying *both* the acquisition and expression of LH have not been studied. This approach naturally allows for the study of the overlapping active populations between acquisition and expression of LH, conferring greater insight into functional changes occurring between these stages.

To label active ensembles with eYFP during LH acquisition, we injected ArcCreER[T2] mice with 4-OHT 5 hr prior to exposure to a series of inescapable footshocks on one side of a shuttle box (IS, inescapable shock group) (*Figure 2A*). A second group of ArcCreER[T2] mice underwent an identical procedure without experiencing footshocks (CT, context trained group). Following 3 days of dark housing, mice underwent 30 trials in which they could escape footshocks by crossing to the opposite chamber of the shuttle box. Ninety minutes following the start of testing, mice were sacrificed, and brains were extracted for investigation of c-Fos expression.

During a 3-min habituation period prior to IS testing, locomotor activity was significantly decreased in the IS group (*Figure 2B*), indicating a fearful association was established with the shuttle box chamber. Across all trials during testing, escape latency was consistently higher in the IS group compared to the CT group (*Figure 2C*), and average latency to cross for trials 11–30 of the testing phase was significantly greater for the IS group compared to the CT group (*Figure 2D*).

After sectioning and immunostaining, we targeted our imaging of eYFP[+] and c-Fos[+] cells (*Figure 2E–H*) across whole coronal sections containing the following regions: hippocampus (HPC), amygdala (AMY), insular cortex (INS), entorhinal cortex (ENT), and raphe nuclei (RAmb). With this semi-targeted mapping approach, we ensured that critical regions previously implicated in the expression of LH and stress responses (*Ineichen et al., 2022*; *Kim et al., 2016*; *Maier and Watkins, 2005*; *Marques et al., 2022*) would be represented in our dataset, along with a myriad of under-investigated regions present within the same atlas plates spread across the rostral–caudal axis. To account for the dendritic/axonal staining pattern of eYFP, a custom algorithm was applied to limit segmentation of

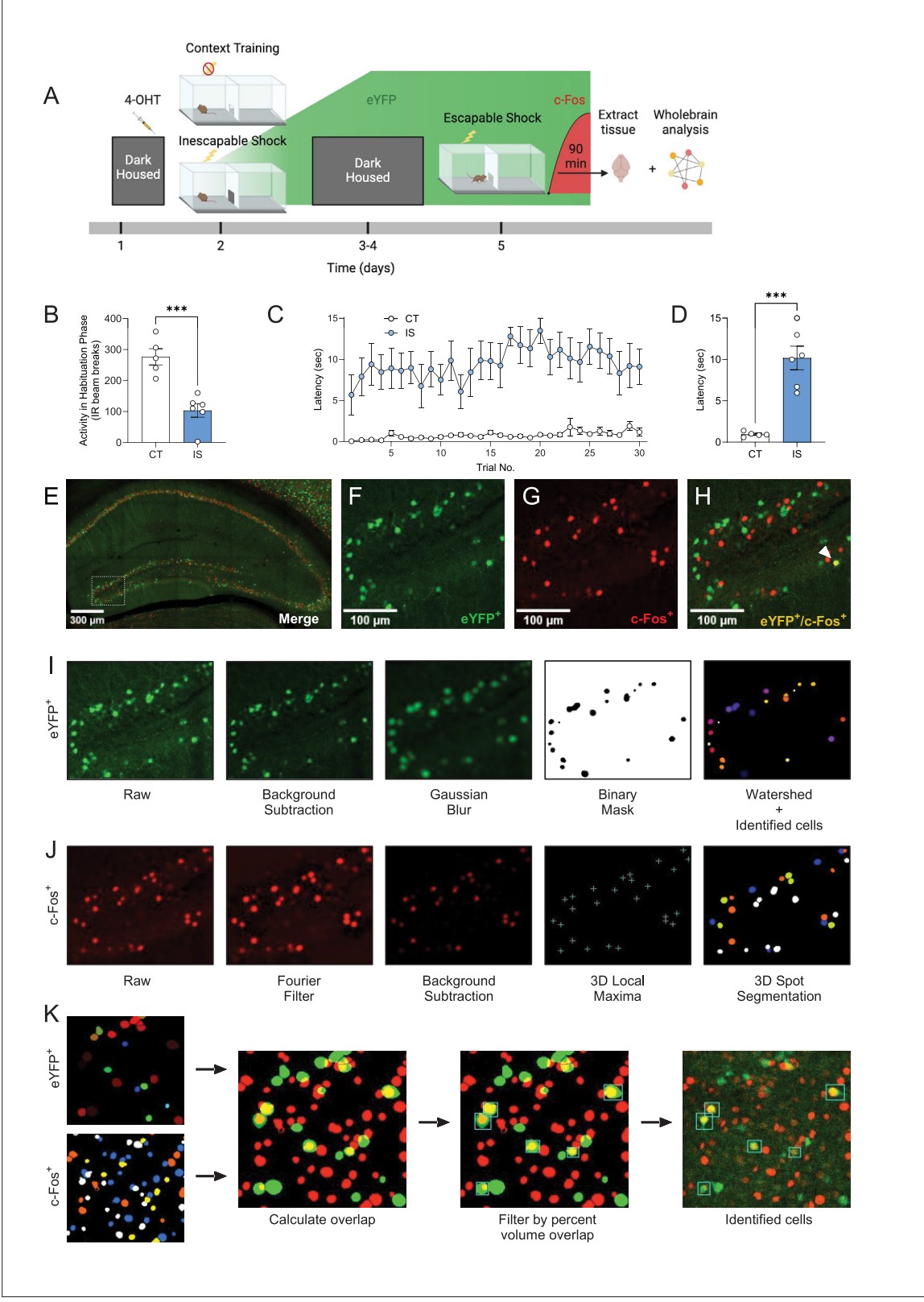

**Figure 2.** Labeling and automatic identification of ensembles active during the acquisition and expression of learned helplessness. (**A**) Experimental design. This panel was created with BioRender. (**B**) Average habituation activity (IR beam breaks) decreases in the shock group, unpaired t-test. (**C**) Prior administration of inescapable shocks (IS) increases subsequent escape latency across 30 trials of shocks. (**D**) Average escape latency across trials 11–30 is higher in the IS group (*n* = 6) compared to the context training (CT) group (*n* = 5), unpaired t-test. (**E**) Representative hippocampal image showing

*Figure 2 continued on next page*

*Figure 2 continued*

eYFP⁺ cells (**F**) and c-Fos⁺ cells (**G**), and their overlap (**H**) in the dDG. (**I**) The consecutive image processing steps optimized for auto-segmentation of eYFP⁺ cells. (**J**) The consecutive image processing steps optimized for auto-segmentation of c-Fos⁺ cells. (**K**) Using the 3D MultiColoc plugin, all possible overlaps are calculated between the segmented eYFP⁺ and c-Fos⁺ objects in ImageJ (middle left). Results are exported and later thresholded by percent volume overlap relative to segmented c-Fos⁺ object (middle right) to identify co-labeled cells (left). ***$p < 0.001$. Error bars represent ± SEM.

The online version of this article includes the following source data and figure supplement(s) for figure 2:

**Source data 1.** Escape latency and beam breaks.

**Figure supplement 1.** Automated segmentation yields comparable results to manual cell counting.

**Figure supplement 2.** Registration projecting segmented cell counts to a standard atlas space.

cellular processes (*Figure 2I*; *Source code 4*). Since this staining pattern is characteristic of other tagging approaches, for example, TRAP1/TRAP2 (*DeNardo et al., 2019*; *Guenthner et al., 2013*), this algorithm is likely generalizable to other activity-dependent tagging mechanisms. Since c-Fos localizes to the cell body, resulting in a punctate filled-spherical staining, a separate optimized algorithm for segmenting this expression pattern was applied (*Figure 2J*; *Source code 3*). Next, all possible segmented object overlaps were compared, and the proportion of volumetric overlap was calculated (relative to the volume of segmented c-Fos⁺ cells). Colocalized cell counts were identified by thresholding by proportion of volume overlap downstream using SMARTTR (*Figure 2K*). The individual image processing steps for both segmentation algorithms and colocalization analysis were all conducted in-batch using ImageJ/Fiji macros, with details extensively outlined in the Methods. These segmentation parameters are adjustable in a GUI menu to flexibly account for a range of resolutions, making them generalizable to other imaging parameters.

Next, we sought to validate the accuracy of our algorithms by comparing the automatically segmented counts of eYFP⁺ and c-Fos⁺ cells to manual cell counts in a subset of images across all mice. Specifically, automated cell counts in the granule cell layer of the DG of the HPC were compared to manual counts from two independent counters. For eYFP⁺, we found that automated segmentation results were highly correlated with manual cell counts across both annotators (*Figure 2—figure supplement 1A–H*; $r = 0.85$ averaged manual counts), and there was high correlation between independent manual counts ($r = 0.92$). Performance was further evaluated by independent quantification of false positive (FP, detected by algorithm alone), false negative (FN, detected by annotator alone), and true positive (TP, detected by both annotator and algorithm) cells on 3333 manual counts from Counter A and 3687 manual counts from Counter B (*Figure 2—figure supplement 1I*). We found that average precision ($P = TP/(TP + FP)$) was 0.92 and average recall ($R = TP/(TP + FN)$) was 0.76. The averaged overall $F1$ score, calculated as the harmonic mean of precision and recall ($1/F1 = (1/P + 1/R)/2$), was 0.85. For c-Fos⁺ cells, we similarly found high correlation between automated cell counts and manual cell counts across both annotators (*Figure 2—figure supplement 1J–Q*; $r = 0.93$ averaged manual counts), as well as high correlation between annotators ($r = 0.93$). We further evaluated the number of FP, FN, and TP cells on 2971 manual counts from Counter A and 3477 manual counts from Counter B (*Figure 2—figure supplement 1R*). The c-Fos-optimized algorithm yielded an average precision of 0.96, an average recall of 0.78, and an average $F1$ score of 0.86. Overall, these results indicate that both algorithms faithfully reflected results that would be obtained from typical manual cell counting approaches and even outperformed metrics reported previously for other automatic segmentation approaches for activity-dependent labeled cells (*Franceschini et al., 2023*).

## Registration and mapping cell counts to a standardized atlas space

We created mouse and slice objects and auto-populated metadata using a custom script that is available on the package website and GitHub repository. We then aligned the preprocessed flattened images (*Figure 2—figure supplement 2A*) to the Allen Mouse Brain atlas templates using the registration() function in SMARTTR. This function interfaces directly with the WholeBrain (*Fürth et al., 2018*) and SMART (*Jin et al., 2022*) packages for interactive user-friendly registration improvement (*Figure 2—figure supplement 2B*). We used functions in SMARTTR to import all raw external segmentation data, including cell counts from both channels as well as co-labeled cell counts (*Figure 2—figure supplement 2C*; c-Fos⁺ cells shown). Segmentation data and registration were next integrated to project counts onto a standardized atlas space (*Figure 2—figure supplement 2D*; c-Fos⁺ cells

shown). Using package functions for data transformation and quality checking, cell counts per region were aggregated across all slice objects per mouse and normalized by volume of the regions mapped. Outliers were cleaned by dropping regional counts greater or less than two standard deviations from their group mean. Only regions represented across both experimental groups per channel were analyzed. We then applied our suite of analysis and visualization functions to examine the brain-wide activity patterns of eYFP+, c-Fos+, and reactivated (co-labeled/eYFP+ cells) cells.

## LH acquisition is marked by enhanced sensory and affective processing

We first investigated neural activity differences (eYFP+ cells) globally and in selected regions during LH acquisition between CT and IS groups. These included targeted isocortical regions (anterior cingulate area, ACA; agranular insula, AI; RSP), dorsal HPC, ventral HPC, and amygdalar subregions (see *Supplementary file 2* for all regional acronyms used). Previous studies have implicated these isocortical regions in LH (*Bauer et al., 2003*; *Kim et al., 2016*), and the ventral HPC and amygdala serve as core limbic centers, especially in fear and aversive processing. Interestingly, during LH acquisition, global brain-wide neural activation levels did not differ between groups (*Figure 3—figure supplement 1A*). Additionally, there were no differences in overall activity among all targeted brain regions between groups (*Figure 3—figure supplement 1B–E*; see *Figure 3—figure supplement 3* for a brain-wide region activation comparison). Correlation of neural activity with behavioral performance in each experimental group was weak across all isocortical, hippocampal, and amygdalar subregions analyzed (*Figure 3—figure supplement 1F–S*), with the exception of the RSP, in which the IS shock group showed strong correlation between eYFP+ cell counts and mean escape latency ($r = 0.88$, $p = 0.02$).

Next, we examined the correlations in IEG expression across brain regions, as strong co-activation or opposing activation can signify functional connectivity between two regions. Correlations were calculated for eYFP+ region counts normalized by volume across all pairs of brain regions for the CT and IS groups, respectively (*Figure 3A, B*). In both networks, there was a higher proportion of positive correlations in cortical regions. However, the distribution of most significant cortical correlations ($p < 0.01$) was predominately in motor and somatosensory areas in the IS group, whereas in the CT group, it was biased toward the auditory areas and integrative or associative regions, such as the ectorhinal cortex (ECT).

To better visualize the most relevant co-activations and analyze their topological structure, we thresholded to retain only the strongest ($r > 0.9$) and most significant ($p < 0.01$) correlations and constructed co-activation networks, in which connections are represented as edges bridging two nodes or brain regions (*Figure 3C*). We then calculated different topology metrics per node, including degree, clustering coefficient, efficiency, and betweenness centrality (*Bullmore and Sporns, 2009*). Degree is the simplest measure of connectivity and represents how many direct connections a single node has. The clustering coefficient represents the tendency of finding a third mutual connection among already connected neighboring nodes (i.e., 'cliquishness'), giving insight into how functionally segregated or integrative a given region is among its neighbors. Efficiency is inversely related to the average path length between a given node and all other nodes in the network, and betweenness indicates how often a particular node lies on the shortest path between two other nodes, which can indicate a region's influence over information flow in functional networks.

We first averaged each of these metrics across all nodes to gain insight into each network's global connectivity patterns (*Figure 3D–H*). While there was no difference in average degree between groups (*Figure 3D*), both networks showed characteristic tailed degree distributions that tend to be found in complex biological systems that follow a power-law distribution (*Figure 3E*; *Barabasi and Albert, 1999*; *Bullmore and Sporns, 2009*; *Bullmore and Sporns, 2012*; *Watts and Strogatz, 1998*). There was also no difference in average clustering coefficient, global efficiency, and mean betweenness centrality (*Figure 3F–H*). As currently, there is no consensus on the ideal approach to choose a threshold to optimize retention of important connections and rejection of spurious ones (*Garrison et al., 2015*), we next examined whether the calculated global metrics might have changed across a wide range of significance thresholds (*Figure 3—figure supplement 2A–D*). Trajectories of degree, clustering coefficient, efficiency, and mean betweenness centrality were qualitatively similar between CT and IS groups across a range of alpha thresholds. Thus, global topology metrics reveal no striking differences in propensity for information transfer between IS and CT networks during LH acquisition.

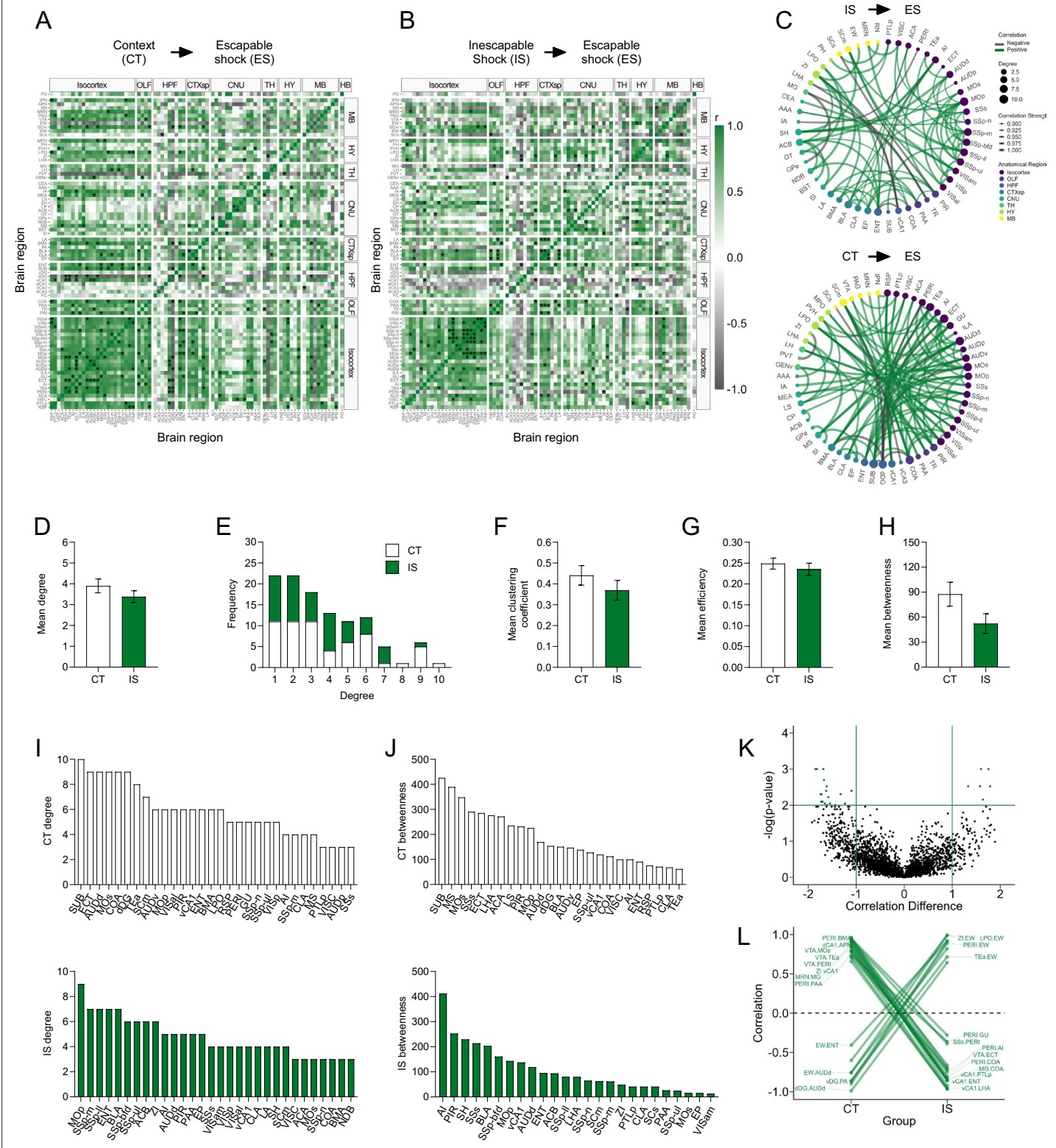

**Figure 3.** Network-level analysis reveals enhanced sensory and affective processing during learned helplessness acquisition. (**A, B**) Regional cross-correlation heatmaps of eYFP⁺ volume normalized cell counts in context trained (CT) and inescapable shock (IS) mice. Significant values are p < 0.01. (**C**) Functional networks constructed after thresholding for the strongest and most significantly correlated or anti-correlated connections (*r* > 0.9, p < 0.01). (**D**) Average degree centrality does not differ between IS and CT groups, unpaired t-test. (**E**) Degree frequency distributions are right-tailed. (**F–H**) Mean clustering coefficient, global efficiency, and mean betweenness centrality do not differ between the CT and IS networks, unpaired t-test. (**I**) The top node

*Figure 3 continued on next page*

*Figure 3 continued*

degree values in descending order for the CT (white, top) and IS (green, bottom) networks indicate which regions are most highly connected. (**J**) The top node betweenness values in descending order for the CT (white, top) and IS (green, bottom) networks indicate which regions are most influential in directing 'information flow'. (**K**) Volcano plot of the Pearson correlation differences ($r_{IS} - r_{CT}$) for all individual regional connections against their p-values calculated from a permutation analysis. Points intersecting or within the upper left or right quadrant represent the regional relationships with the greatest change (|correlation difference| >1), that were most significant (p < 0.01). (**L**) A parallel coordinate plot highlighting individual significant changed regional correlations between groups, as well as the direction of their change. Error bars represent ± SEM.

The online version of this article includes the following figure supplement(s) for figure 3:

**Figure supplement 1.** Activity is not heightened globally or across targeted subregions following exposure to inescapable shock (IS).

**Figure supplement 2.** Global network topology metrics during exposure to inescapable shock across a range of thresholds.

**Figure supplement 3.** Differential expression of eYFP activity across all mapped regions.

To better investigate which individual regions are most differentially influential between networks, we looked specifically at the distribution of individual nodal degree centrality and betweenness centrality values (*Figure 3I–J*). The degree distributions revealed that many primary somatosensory (SSp) regions (mouth, SSp-m; lower limb, SSp-ll; barrel field, SSp-bfd; upper limb, SSp-ul), motor regions (primary motor area, MOp), and limbic regions, including the nucleus accumbens (ACB) and BLA, were highly connected in the IS. Comparatively, far fewer sensory regions showed such high degree in the CT networks. Despite the high connectivity among mostly sensory regions, IS network betweenness distributions showed that additional key structures such as the AI, piriform cortex (PIR), septohippocampal nucleus (SH), and ENT act as influential information bridges in the IS network. The AI is a critical region involved in pain processing (*Labrakakis, 2023*), and its central network engagement is largely consistent with experiencing a painful sensory event. In contrast, structures such as the subiculum (SUB), ECT, dorsal dentate gyrus (dDG), and medial septal nucleus (MS) were most influential in CT networks, and betweenness distributions also reveal that the SUB was most influential. The subiculum is well known to play a role in spatial representation during exploration (*Matsumoto et al., 2019*; *O'Mara et al., 2009*), and the MS and dDG are, respectively, involved in motivated locomotion (*Mocellin and Mikulovic, 2021*) and contextual processing (*Tuncdemir et al., 2019*). The overall patterns seen in the individual regional degree and betweenness distributions indicate a strong engagement of areas involved in contextual and navigational processing in the CT network, whereas there appears to be greater influences of regions involved in sensory and affective processing of shock (e.g., BLA and AI), and motor reactivity (e.g., MOp) in the IS network.

## LH acquisition alters ventral CA1, ventral tegmental, and perirhinal connections

To better understand how coordinated activity among individual connections differs between groups, we performed a permutation analysis. Regional correlations from the CT group were subtracted from corresponding correlations in the IS group and compared against a permuted null distribution. We further analyzed only correlations differences that were large (r difference ≥1) and highly significant (p < 0.01) (*Figure 3K*). A number of interesting patterns emerged from this analysis. Specifically, connections involving either the ventral CA1 (vCA1), ventral tegmental area (VTA), or perirhinal cortex (PERI) were strongly positively correlated in the CT group, whereas they became negatively correlated in the IS group (*Figure 3L*). This pattern was consistent among connections between the vCA1 with associative areas (posterior parietal association areas, PTLp; ENT) and hypothalamic areas (zona incerta, ZI; lateral hypothalamic area, LHA), and between the PERI with primarily limbic, especially amygdalar regions (AI, BLA, piriform amygdalar area, and cortical amygdalar area), and the VTA with associative regions (temporal association areas, TEa; ECT, PERI). The VTA is centrally involved in reward processing and functional inversion of its connections may be indicative of the aversive experiencing of inescapable footshocks. Additionally, the vCA1 is known to be involved in processing of affective states and fear retrieval, with the vCA1–LHA projection in particular being enriched in representations of anxiety (*Jimenez et al., 2018*; *Jimenez et al., 2020*), suggesting functional alterations of this particular connection may be important in the development of LH. Finally, the PERI region is involved in object perception and sensory feature processing (*Murray and Richmond, 2001*), and thus, its

strongly altered connectivity may indicate differences in how sensory features of the environment are assigned an affective value in IS versus CT mice.

## LH expression is characterized by a reversal in functional connectivity of the substantia nigra

We applied the same analysis approach to the mapped c-Fos[+] dataset, representing cells active during the expression of LH, when mice are tested under ES conditions. Global brain-wide neural activation levels during LH expression did not differ between groups (*Figure 4—figure supplement 1A*). Additionally, in the ACA, AI, RSP, dorsal HPC, ventral HPC, and amygdalar subregions (*Figure 4—figure supplement 1B–E*), there were no differences in overall activity between groups. Correlation of activity with behavioral performance in each experimental group was weak across all isocortical, HPC, and amygdalar subregions analyzed (*Figure 4—figure supplement 1F–S*). Thus, examining only absolute activation patterns did not reveal obvious neural signatures of expression of LH (see *Figure 4—figure supplement 3* for a brain-wide region activation comparison).

We next examined the functional network signatures. As was performed for eYFP[+] networks, we calculated regional correlations (*Figure 4A, B*), and thresholded to retain only the strongest (*r* > 0.9) and most significant (p < 0.005) correlations to construct networks (*Figure 4C*). Qualitatively, we observed that somatosensory and motor areas appear more integrated with amygdalar and hippocampal structures in CT compared to IS networks (*Figure 4C*). We then calculated the same global network topology metrics previously described. We found that there was no mean difference in degree (*Figure 4—figure supplement 2A*), although both networks showed the expected tailed degree distribution (*Figure 4—figure supplement 2B*). There was also no difference in clustering coefficient, global efficiency, and mean betweenness centrality (*Figure 4—figure supplement 2C–E*). We validated the stable nature of this observation by recalculating these global metrics across a wide range of possible significance thresholds and observed similar trajectories between CT and IS networks across all metrics (*Figure 4—figure supplement 2F–I*). Similarly, to our eYFP[+] analysis, these findings indicate that the expression of LH is not characterized by global differences in efficacy of information transfer in functional networks.

To investigate which key influential regions differ between each network, we plotted the individual region degree and betweenness distributions for CT and IS groups (*Figure 4D, E*). The most highly interconnected region in the CT network was the nucleus reunions (RE), and the region with the highest betweenness value was the diagonal band nucleus (NDB). In contrast, for the IS network, the most highly interconnected regions include the dCA1 and ventral medial nucleus of the thalamus (VM), whereas two basal ganglia structures, the subthalamic nucleus (STN) and the substantia nigra, reticular part (SNr), displayed the highest betweenness centrality. To further investigate how regions showing either the highest degree or betweenness centrality in the IS group are functionally changed, we plotted the distribution of Pearson correlation coefficients for all dCA1, VM, STN, and SNr connections (*Figure 4F–I*). While there were no differences in the correlation distributions of the dCA1, VM, and STN, in the SNr, there was a strong shift from predominantly negative correlations in the CT group to positive correlations in the IS group (*Figure 4I*). Altogether, these findings indicate that the VM, a central thalamic relay structure which integrates limbic and prefrontal cortical signals to guide motor behavior (*Anastasiades et al., 2021*; *Monconduit et al., 1999*; *Sieveritz et al., 2019*), and the NDB, a basal forebrain structure capable of mediating motivated movement (*Mocellin and Mikulovic, 2021*), may be involved in the cognitive processes involved in generating adaptive escape responses in control animals. General reversal of functional connectivity in the substantia nigra (SN) when comparing the CT and IS suggests that the SN may be a critical region involved in gating of motor processing involved in an adaptive crossing response.

Finally, we performed a permutation analysis as previously described to examine pairwise regional correlation changes between groups (*Figure 4J, K*). In line with our previous findings, we found a large proportion of the significantly altered connections in the positive direction in the IS group relative to the CT included the SNr. These SNr connections involved a number of sensory, motor, and visual processing regions. Conversely, a number of positively correlated connections involving the PERI region in the CT group shifted to negative correlations in the IS group. Among these connections were various midbrain structures (VTA and MRN) and visuomotor processing centers (SCm), and hypothalamic structures (ZI). As strong correlation reversals of PERI connections between CT and IS groups

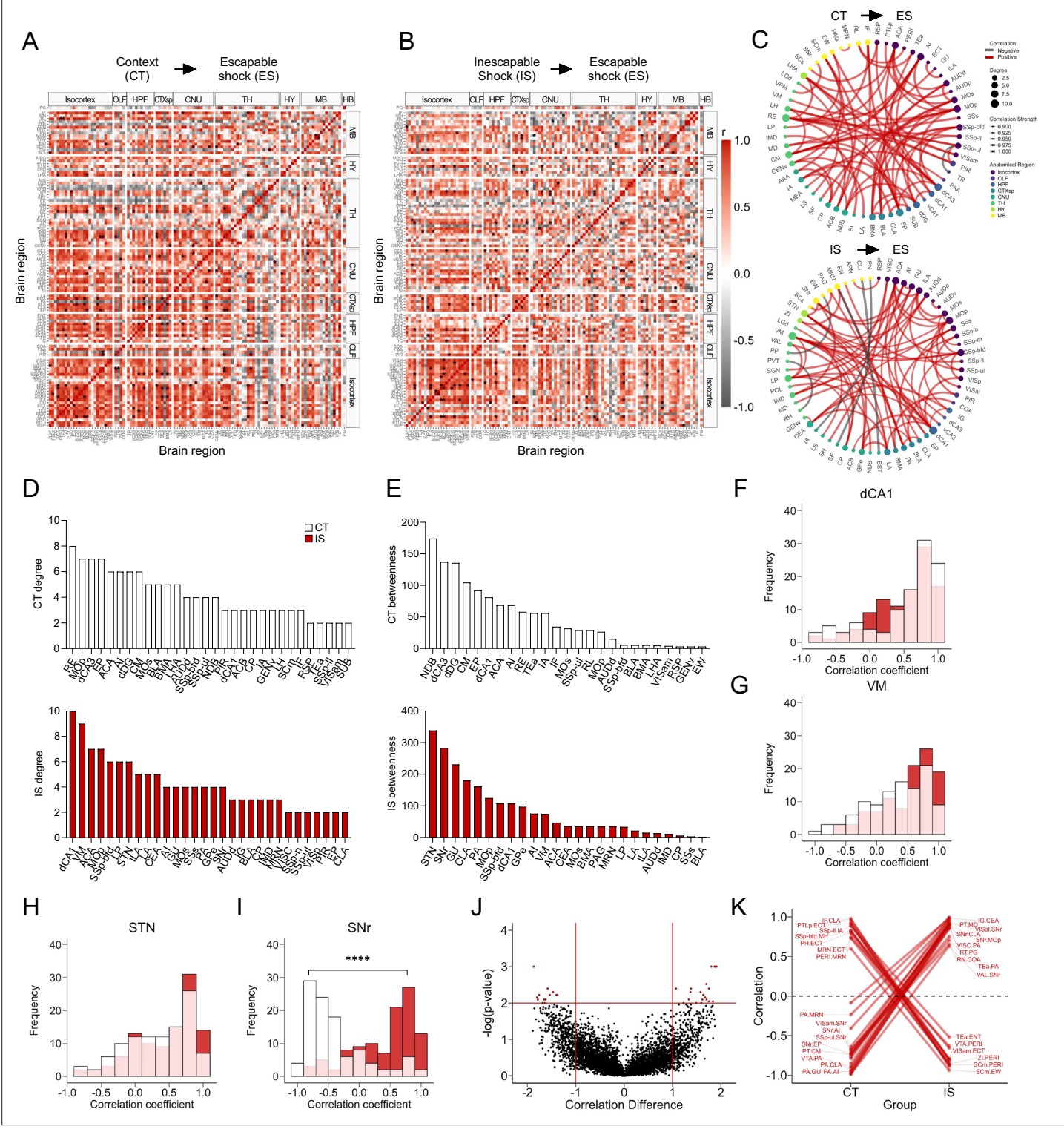

**Figure 4.** Network-level analysis reveals influential altered functional connectivity of the substantia nigra during learned helplessness expression. (**A, B**) Regional cross-correlation heatmaps of c-Fos⁺ expression in context trained (CT) and inescapable shock (IS) groups. Significant values are p < 0.005. (**C**) Functional networks constructed after thresholding for the strongest and most significantly correlated or anti-correlated connections (*r* > 0.9, p < 0.005). (**D**) Highest individual node degree values in descending order for the CT (white, top) and IS (red, bottom) networks indicate which regions are most highly connected. (**E**) Node betweenness values in descending order for the CT (white, top) and IS (red, bottom) networks. (**F–I**) Pearson correlation distributions of the dorsal CA1 (dCA1), ventral medial nucleus of the thalamus (VM), subthalamic nucleus (STN), and substantia nigra, reticular part (SNr). Distributions between CT and IS groups significantly differ in the SNr, two-sample Kolmogorov-Smirnov test. (**J–K**) Volcano plot and parallel coordinate

*Figure 4 continued on next page*

*Figure 4 continued*

plots highlighting the permuted correlation differences ($r_{IS} - r_{CT}$) that show the greatest change (|correlation difference| >1), and are most significant (p < 0.01) between the CT and IS groups. ****p < 0.0001. Error bars represent ± SEM.

The online version of this article includes the following figure supplement(s) for figure 4:

**Figure supplement 1.** Absolute activity of targeted isocortical, hippocampal, and amygdala regions is not associated with learned helpless expression.

**Figure supplement 2.** Properties of networks constructed from brain-wide c-Fos$^+$ regional co-expression.

**Figure supplement 3.** Differential expression of c-Fos activity across all mapped regions.

were similarly seen in ensembles active during LH acquisition, early modulation of PERI function may be an early functional signature of development of LH.

## LH is characterized by dampened reactivation of cells previously active during IS

We next leveraged the capability of the SMARTTR package to study co-labeled eYFP$^+$/c-Fos$^+$ cell populations active during both acquisition and expression of LH. To investigate the proportion of co-labeled cells reactivated among the original ensemble active during IS acquisition or CT, co-labeled cells were divided by eYFP$^+$ counts to generate reactivation proportions (co-labeled/eYFP$^+$ cells). We applied the same analysis approach to the proportion of reactivated cells, as was performed for c-Fos$^+$ and eYFP$^+$ networks.

Targeted regional analysis of reactivation activity (*Figure 5A–E*) revealed that a number of regions, including the ACA, AI, dDG, dCA3, and MEA, were significantly decreased in their reactivation proportions (*Figure 5F–I*). Interestingly, a general analysis of global reactivation proportions shows a trending brain-wide decrease in reactivated activity (p = 0.0560, *Figure 5—figure supplement 1A*; see *Figure 5—figure supplement 3* for a brain-wide regional comparisons). Additionally, we found that reactivated activity in LA, a central region involved in associative fear learning (*Johansen et al., 2010*), showed a strong correlation with mean latency to cross in the IS group (*Figure 5J*). An additional exploratory analysis of reactivation activity as a proportion of the ensembles active during ES (co-labeled/c-Fos$^+$ cells) revealed a similar trending global brain-wide decrease in reactivation proportions (*Figure 5—figure supplement 4A* see *Figure 5—figure supplement 6* for a brain-wide regional comparisons). Overall, these findings indicate that a general global dampening of reactivation activity may be a novel marker of the traumatized helpless state.

## Altered functioning of the insula and central amygdala in reactivated cells characterizes the LH state

We next investigated functional networks of reactivation activity by plotting regional correlations (*Figure 5K, L*), thresholding to retain the strongest and most significant correlations (r > 0.9, p < 0.01), and constructing networks (*Figure 5M*). Similar to eYFP and c-Fos networks, examination of global topology metrics revealed no significant differences in degree, clustering coefficient, efficiency, and betweenness centrality (*Figure 5—figure supplement 2A–E*), and we confirmed the stability of this observation by seeing similar trajectories of these metrics between groups across a wide range of significance thresholds (*Figure 5—figure supplement 2F–I*).

We next plotted the individual region degree and betweenness distributions (*Figure 5N, O*) and found that the most highly interconnected regions in the IS network were the dCA1, AI, secondary motor area (MOs), and the primary somatosensory area, nose (SSp-n). The regions with the highest betweenness centrality were the ACA and MOs. For this reason, we further plotted the distribution of all Pearson correlation coefficients for the dCA1, AI, ACA, and MOs (*Figure 5P-S*). While there were no differences found in the correlation distributions for the dCA1, ACA, and MOs, in the AI, there was a significant increase in the proportion of positive correlations in the IS group compared to the CT group (*Figure 5Q*), indicating a general strengthening of functional connectivity of the AI among reactivated cells. In an exploratory analysis of co-labeled/c-Fos$^+$ cell activity, we also found that AI correlations to the vDG region were preferentially strengthened in the IS group (*Figure 5—figure supplement 5L*).

Finally, we used a permutation analysis, as done previously (*Figure 5T–U*) to assess regional correlations differences that were large (r difference ≥1) and highly significant (p < 0.01) between the two

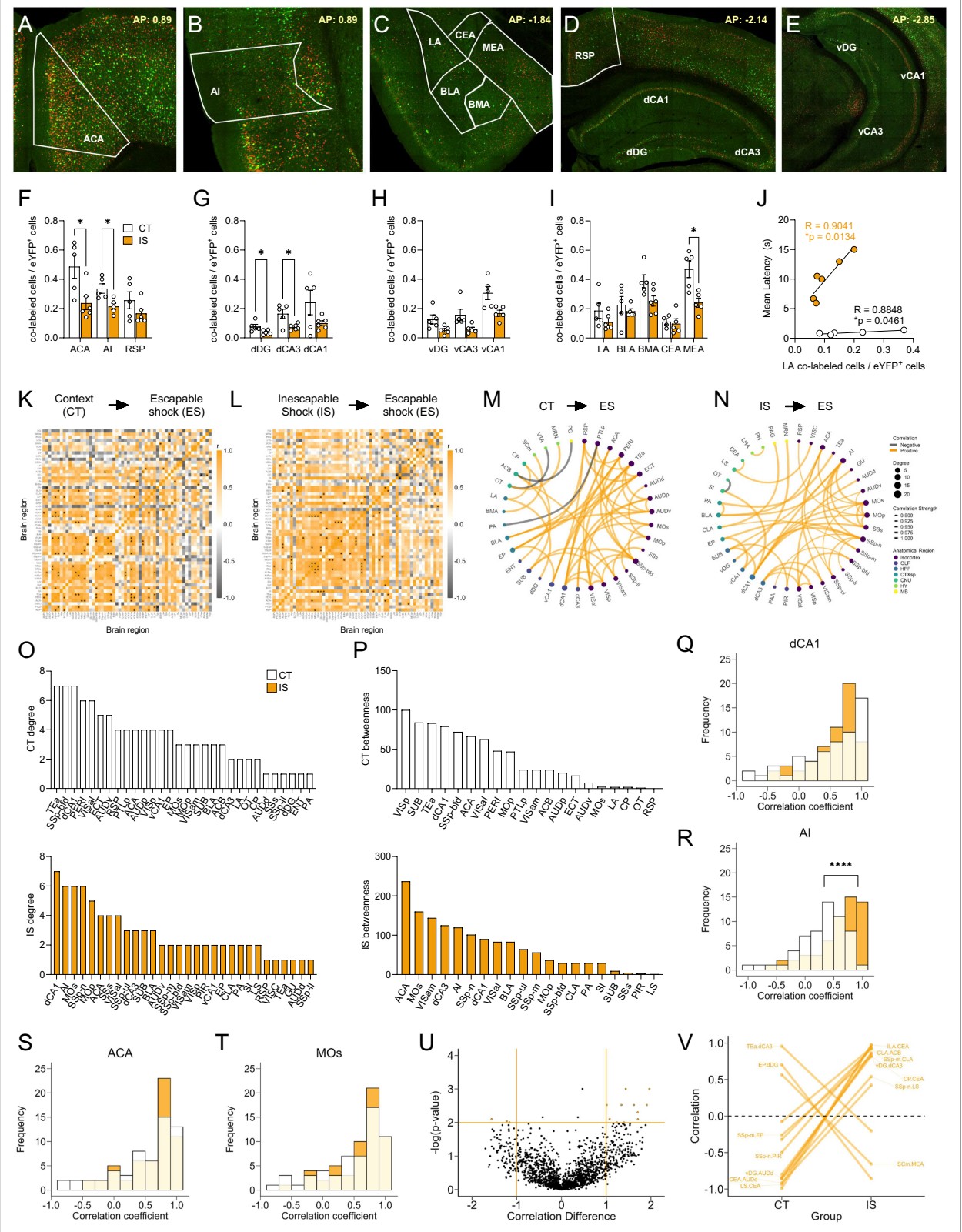

**Figure 5.** Network analysis of reactivated inescapable shock (IS) ensembles during learned helplessness reveals altered functional connectivity. (**A–E**) Representative images of regions identified for targeted analysis, including isocortical regions (anterior cingulate area, ACA; agranular insula, AI; and retrosplenial area, RSP), dorsal hippocampal regions (dorsal dentate gyrus, dDG; dorsal CA3, dCA3; and dorsal CA1, dCA1), ventral hippocampal regions (ventral dentate gyrus, vDG; ventral CA3, vCA3; and ventral CA1, vCA1), and amygdalar areas (lateral amygdalar nucleus, LA; basolateral

*Figure 5 continued on next page*

*Figure 5 continued*

amygdalar nucleus, BLA; basomedial amygdalar nucleus, BMA; central amygdalar nucleus, and CEA; medial amygdalar nucleus, MEA). Representative region overlays were manually drawn. (**F–I**) The ACA, AI, dDG, dCA3, and MEA show significantly decreased reactivation activity (co-labeled cells/eYFP⁺ cells) in the mice exposed to IS compared to context training, multiple t-test (Holm-Sidak method for multiple comparisons). (**J**) LA reactivation activity in both context trained (CT, n=5) and IS mice (n=6) shows positive correlation to escape latency. (**K, L**) Correlation heatmaps of reactivation activity in CT and IS mice. (**M, N**) Functional networks constructed after thresholding for the strongest and most significant correlated or anti-correlated connections ($r > 0.9$, $p < 0.01$). (**O**) Top individual node degree values in descending for the CT (white, top) and IS (yellow, bottom) networks. (**P**) Highest node betweenness values in descending for the CT (white, top) and IS (yellow, bottom) networks. (**Q–T**) Pearson correlation distributions of the dCA1, AI, ACA, secondary motor area (MOs), and substantia nigra, reticular part (SNr). Distributions between CT and IS groups significantly differ in the AI, two-sample Kolmogorov-Smirnov test. (**U,V**) Volcano plot and parallel coordinate plots highlighting the permuted correlation differences ($r_{IS} - r_{CT}$) of functional connections of reactivated activity showing the greatest change (|correlation difference| >1), and are most significant ($p < 0.01$) between the CT and IS groups. *$p < 0.05$, ****$p < 0.0001$. Error bars represent ± SEM.

The online version of this article includes the following figure supplement(s) for figure 5:

**Figure supplement 1.** Global reactivation activity during escapable shock is dampened in mice previously exposed to inescapable shock (IS).

**Figure supplement 2.** Properties of networks constructed from functional connections between reactivated ensembles (co-labeled/eYFP⁺ cells).

**Figure supplement 3.** Differential reactivation proportions across all mapped regions.

**Figure supplement 4.** Reactivated activity proportions are decreased in targeted ventral hippocampal regions following inescapable shock (IS).

**Figure supplement 5.** Supplementary network-level analysis of co-labeled/c-Fos⁺ activity reveals high sensory functional connectivity.

**Figure supplement 6.** Differential co-labeled/c-Fos⁺ proportions across all mapped regions.

groups. Interestingly, a pattern emerged whereby there was a disproportionate number of strongly negative connections in the CT group that shifted to strongly positive connections in the IS group, many of which involved the CEA, a primary fear and pain processing center. These significantly altered CEA connections included regions such as the infralimbic area, which is involved in flexible response inhibition and decision-making (*Barker et al., 2014*), the caudoputamen, a basal ganglia structure involved in motor regulation, and the lateral septum, which is involved in emotional regulation and stress responding (*Wirtshafter and Wilson, 2021*; *Yadin et al., 1993*). Altogether, these observations suggest that a functional change in the AI and strong functional reversal of key CEA connections in reactivated neural populations may underlie the process of developing of LH.

## Discussion

Activity-dependent tagging systems have been invaluable for studies investigating the shared overlapping ensembles active across two separate experiences. Such studies have revealed invaluable insights on the ensembles responsible for memory storage and retrieval (i.e., engrams). However, most of these studies have relied on targeted investigation of single or few regions (*Cowansage et al., 2014*; *Denny et al., 2014*). The few publications reporting mapping of brain-wide tagged ensembles have relied on intact tissue clearing, 3D imaging, and registration workflows that, while advantageous in their ability to keep tissue intact, are specialized and difficult to implement widely, and/or rely on commercial software (e.g., Imaris) (*Franceschini et al., 2023*; *Roy et al., 2022*).

To our knowledge, only one paper has reported mapping and analysis of colocalized ensembles without introducing a generalizable workflow (*Roy et al., 2022*). Thus far, there have been scant accessible approaches that allow for the scalable brain-wide mapping of multiple ensembles as well as their overlap in traditionally immunolabeled coronal sections. Moreover, there are no standardized open-source tools to analyze functional connectivity signatures based on network analysis of brain-wide IEG activity patterns.

Here, we present our workflow for automated segmentation optimized for the identification of eYFP⁺ cells (indicative of previously Arc⁺ cells), c-Fos⁺ cells, and their co-labeled overlap. This process is open-source, as it is conducted entirely in ImageJ/FIJI (*Schindelin et al., 2012*), and relies on existing image processing toolkits that are freely available. Moreover, we introduce the SMARTTR package which offers a standardized API for the importation of segmentation output from our workflow, and for atlas registration and mapping by interfacing with previously published R packages (*Fürth et al., 2018*; *Jin et al., 2022*). However, perhaps the most useful and novel feature of SMARTTR is the inclusion of numerous data management, quality checking, graph theory analyses, and visualization functions that are user-friendly, requiring only rudimentary programming knowledge.

We demonstrated the utility of our workflow by applying it to map two ensemble populations active during the experience of IS (or context) or later during exposure to ES. Our goal was to gain insight into how functional activity networks are changed during both LH acquisition and expression, which has never previously been studied. As the LH paradigm models a fundamental symptomology of trauma-related depression and anxiety states, this work has the major clinical implication of improving our understanding of how these states emerge following trauma. Our analysis of global network properties of eYFP$^+$, c-Fos$^+$, and co-labeled/eYFP$^+$ activity expression revealed no differences in topological metrics, suggesting that the learned helpless state is not characterized by gross deficits or alterations in information transfer during acquisition or expression of LH.

However, our region and connection specific analyses of eYFP$^+$ ensemble networks suggest that, when experiencing IS compared to context exposure, there is a functional shift from emphasis on navigational and contextual processing to sensory and affective processing. Network-wise, this appears to be represented by a decreased influence of regions such as the subiculum, which is centrally involved with spatial navigation (*Matsumoto et al., 2019*; *O'Mara et al., 2009*), to generally high interconnectivity of somatosensory regions, and those potentially involved with affective processing of shock, such as the BLA and AI.

Surprisingly, we did not find strong evidence of absolute activity differences between groups across a targeted or brain-wide regional comparisons, and overall amygdalar activity in IS mice was comparable to that of CT mice during LH training. This is in contrast to previous findings showing *Arc* gene expression in the amygdala is upregulated 2 hr after IS (*Machida et al., 2018*). However, a key methodological difference in our approach is that the ArcCreER$^{T2}$ × eYFP tagging strategy labels previously Arc-expressing cells with eYFP over a longer window of time, as the serum half-life of 4-OHT is approximately 6 hr (*Cazzulino et al., 2016*). Thus, neurons active during consolidation of the experience in the hours following acute shock or context exploration may also be recruited into the tagged ensemble.

During ES testing, network analysis revealed surprising high influence of two basal ganglia structures, the STN and the SNr, and that there was a striking functional reversal of the SNr between the CT and IS groups. Interestingly, recent work has shown that stressor controllability can change intrinsic dynamics and electrophysiological properties of SN neurons (*Yao et al., 2021*). These findings, in conjunction with our results, suggest the possibility that the SN activity is altered by loss of control over a stressor in a manner that may gate an adaptive motor response such as crossing.

Finally, we examined the overlapping ensembles active during both LH acquisition and expression (eYFP$^+$/c-Fos$^+$) as a proportion of the originally active ensemble population (eYFP$^+$). Interestingly, we found a robust decrease in reactivation activity in several targeted brain regions, alongside a trending brain-wide decrease. To our knowledge, this pattern of widescale decreased reactivity following exposure to a traumatic event has not been described before and suggests a potential phenomenon of neural plasticity which acts to suppress future ensemble reactivity as a major part of the neurobiology underlying LH. We also found a general correlative association between LA co-labeled activity and escape latency, suggesting reactivation of LA ensembles initially active during IS involved in regulating response to ES.

Performing network analyses, we identified the AI as a highly interconnected region in the IS networks, which showed a significantly strengthened distribution of functional connections compared to the CT group. As an integrative hub for sensory, affective, and cognitive function, in addition to its role in pain processing, the insula is involved in regulating fear and anxiety behaviors (*Casanova et al., 2016*; *Labrakakis, 2023*; *Nicolas et al., 2023*; *Rogers-Carter et al., 2018*). The influential positioning of the AI in the IS networks, along with its functional alteration, indicates that reactivated functional activity in the AI may be critical in mediating the helpless state. Network analysis also revealed the ACA as another potential key region, as the ACA displayed the highest betweenness centrality in the IS network. This finding is consistent with a previous study of long-term fear memory using network analysis of c-Fos expression which also found the ACA as an important hub within the fear network (*Wheeler et al., 2013*). Finally, our permutation analysis revealed a number of key CEA connections that were strongly functionally reversed between IS and CT groups, suggesting possible functional targets for future investigation. Given these many novel insights, we believe the strength of SMARTTR and our multiple-ensemble mapping workflow lies in its use as a hypothesis-generating toolkit, which

may lead to new questions and suggest future mechanistic studies that may draw more definitive causal conclusions.

## Methodological considerations

There are many factors to consider regarding the choice of our workflow over other options. While serial coronal sectioning is not as rapid as intact tissue immunostaining, this approach allows for adequate antibody penetration for two or more ensemble populations in deep brain structures for high signal-to-noise staining. Intact whole-brain immunohistochemistry, typically in combination with lipid clearing (*Chung et al., 2013*; *Park et al., 2019*; *Renier et al., 2014*), has been demonstrated to label *single* ensembles well. However, insufficient penetration of multiple antibodies or purging of epitopes, especially during permeabilization steps in hydrophobic clearing approaches, remains a major technical barrier to widespread adoption in mapping engrams. We have found many of these methods result in confounding staining gradients from the surface to deeper depths of the tissue (*Pavlova et al., 2018*), leading to inaccurate visualization of multiple ensembles in structures such as the hippocampus, which confounds study of biologically relevant overlapping populations. Applying intact tissue clearing approaches to brain-wide engram mapping remains a tremendous technical hurdle for most.

A second limitation of this workflow is that the accuracy of registration alignment depends on the quality of physical sectioning, user annotation, and correction of registrations. Furthermore, accurate registration in the *z*-plane is limited to a resolution of ~100 μm. This is because the registration capabilities of WholeBrain (*Fürth et al., 2018*) and SMART (*Jin et al., 2022*), the underlying packages used for registration and mapping, require coronal sections to align closely to the sectioning plane of the Allen Mouse Brain Atlas. Thus, registration alignment requires both technical skill and familiarity with brain-wide anatomy. However, this challenge can be mitigated with the use of brain matrices, which can help mount an even cutting plane during physical sectioning. Additionally, limited registration accuracy along the z-plane is still comparable to other recent volumetric registration approaches reporting accuracies of ~300 μm (*Franceschini et al., 2023*), whereas maximal registration accuracy along the *x*- and *y*-planes is much higher due to WholeBrain's scale-invariant spline-based atlas (*Fürth et al., 2018*).

## Future directions

There are a number of alternative mapping approaches that can account for sectioning angle during registration (*Puchades et al., 2019*; *Song et al., 2020*). The structure of SMARTTR is modular such that additional pipeline entry points are possible if separate workflows are used for segmentation, registration, or mapping. For example, independently mapped data can be easily imported for integration with SMARTTR's network analysis functions. Similarly, with some simple programming, it is possible to import segmented data from other approaches, such as CellPose (*Stringer et al., 2021*) or QuPath (*Bankhead et al., 2017*), and still take advantage of the built-in registration, mapping, and analysis capabilities in SMARTTR. We anticipate, based on community feedback, that additional analysis features will be continually added to the package. For example, since the default supported atlas ontology is the Allen Mouse Brain Common Coordinate Framework, the merged Franklin–Paxinos/CCF atlas nomenclature (*Chon et al., 2019*) was incorporated to analysis and visualization functions based on collaborator feedback. Finally, while this pipeline was designed to be especially user-friendly to novice programmers, future iterations may be even easier to learn through incorporation of a GUI menu for easy creation and modification of object metadata, running analyses, and visualizations.

## Materials and methods

**Key resources table**

| Reagent type (species) or resource | Designation | Source or reference | Identifiers | Additional information |
|---|---|---|---|---|
| Strain, strain background (mouse, male) | 129S6/SvEv | Taconic | RRID:MGI:3044417 129S6/SvEvTac | |

*Continued on next page*

*Continued*

| Reagent type (species) or resource | Designation | Source or reference | Identifiers | Additional information |
|---|---|---|---|---|
| Genetic reagent (mouse) | ArcCreERT2(+) × eYFP | *Denny et al., 2014*; *Srinivas et al., 2001* | RRID:IMSR_JAX:039977 (Stock no. 039977); RRID:IMSR_JAX:006148 (Stock no. 006148) | Bred in-house on 129S6/SvEv background |
| Antibody | anti-GFP (chicken polyclonal) | Abcam, Cambridge, MA | RRID:AB_2936447, Cat.#ab13970 | IF (1:2000) |
| Antibody | IgG anti-c-Fos (Rabbit polyclonal) | SySy, Goettingen, Germany | RRID:AB_2231974, Cat.#226 003 | IF (1:5000) |
| Antibody | Alexa 647 (conjugated Donkey Anti-Rabbit IgG) | Life Technologies, Carlsbad, CA | RRID:AB_2536183, Cat.#A-31573 | IF (1:500) |
| Antibody | Cy2 (conjugated Donkey Anti-Chicken IgG) | Jackson ImmunoResearch, West Grove, PA | RRID:AB_2340370, Cat.#703-225-155 | IF (1:250) |
| Chemical compound, drug | 4-Hydroxytamoxifen | Sigma-Aldrich, St. Louis, MO | Cat.#H6278 | |
| Software, algorithm | ImageJ | http://fiji.sc/ | | (v.1.52p) |
| Software, algorithm | R | https://www.r-project.org/ | | (v 3.6.3) |
| Software, algorithm | RStudio | https://posit.co/download/rstudio-desktop/ | | (2022.07.2) |
| Software, algorithm | SMARTTR | https://github.com/mjin1812/SMARTTR, copy archived at *Jin and Ogundare, 2025* | | Maintainer: Michelle Jin (mj2947@cumc.columbia.edu) |
| Other | Fluoromount G | Electron Microscopy Sciences, Hatfield, PA | RRID:AB_2572296, Cat.#17984-25 | See Immunohistochemistry section |

## Mice

ArcCreER^T2 (*Denny et al., 2014*) × R26R-STOP-floxed-eYFP homozygous female mice were bred with R26R-STOP-floxed-eYFP homozygous male mice (*Srinivas et al., 2001*). All experimental mice were ArcCreER^T2 (+) and homozygous for the eYFP reporter. ArcCreER^T2 (+) × eYFP mice are on a 129S6/SvEv background, as they have been backcrossed for more than 10 generations onto a 129S6/SvEv line. The age of the mice utilized for experimentation ranged from 6 to 7 months.

## Learned helplessness

### 4-Hydroxytamoxifen

Recombination was induced with 4-OHT (Sigma, St Louis, MO). 4-OHT was dissolved by sonication in 10% EtOH/90% corn oil at a concentration of 10 mg/ml. One injection of 200 µl (2 mg) was intraperitoneally administered to each mouse.

LH was administered as previously described (*Brachman et al., 2016*). Briefly, in the IS phase, mice in the IS group were habituated for 3 min to one side of a two-chamber shuttle box (model ENV 010MD; Med Associates, St. Albans, VT) containing a grid floor connected to a scrambled shock generator (model ENV 414S, Med Associates). The shuttle box was equipped with eight infrared beams (IR, four on each side) for detecting position and activity of the animal, and an automated guillotine door separating the chambers. Training consisted of shocks, each with a 3-s average duration, at 0.5 mA, and with an intertrial interval (ITI) of approximately 15 s. Mice in the CT group were exposed to the shuttle box for an identical duration of time, without receiving shocks.

In the ES phase, all mice were tested in the same shuttle box in which they were trained. Each mouse was placed into the left chamber with the door raised and was allowed to freely explore both chambers for 3 min. IR beam breaks were used as a measure of locomotor activity during this habituation phase. The door then closed automatically, and 30 trials were administered. At the beginning of each trial, the door was raised and 5 s later, a foot shock (0.5 mA) was delivered. The subject's exit from the shocked side ended the trial. If the mouse did not exit after 15 s, the shock was turned off

and the trial ended. The door was lowered at the end of the trial. A session consisted of 30 trials, each separated by a 30-s ITI. Escape latencies were computed as the time from shock onset to the end of trial. If the subject failed to make a transition, the maximum 15 s was used for the escape latency score. Mean latencies per subject were calculated based on performance across trials 11–30. Statistical analysis results are listed in *Supplementary file 1*.

## Memory trace tagging

Five hours before IS or CT, mice were injected with 200 µl (2 mg) of 4-OHT intraperitoneally. Following behavioral training, mice were dark housed for 3 continuous days (days 2–4). Mice were taken out of the dark on the morning of day 4, cages were changed, and they were returned to the normal colony room prior to behavioral testing. All precautions to prevent disturbances to the ArcCreER$^{T2}$ × eYFP mice during dark housing were taken to reduce off-target labeling. Following ES testing, mice were euthanized 90 min after the start of the test to allow for visualization of c-Fos expression.

## Statistical analysis

Behavioral data and summary neural data were analyzed with Prism v.9.0 or v.10.0 using an alpha value of 0.05. The behavioral effects of IS were analyzed using a *t*-test on averaged latencies between trials 11–30 and, using a two-way ANOVA to assess the effects of Group, Trial, and the Group × Trial interaction, where appropriate. All statistical tests and p-values for behavioral data are listed in *Supplementary file 1*. For comparison of regional Pearson correlation value distributions between groups, a two-sample Kolmogorov–Smirnov test was used. For comparisons of neural activation levels across targeted brain regions, multiple *t*-tests were conducted with a Holm–Sidak correction for multiple comparisons. All statistical tests and p-values for summary neural data are reported in *Supplementary file 3*.

## Immunohistochemistry

Mice were deeply anesthetized and transcardially perfused as previously described (*Denny et al., 2012*). Brains were extracted and postfixed overnight in 4% paraformaldehyde at 4°C, and cryoprotected in 30% sucrose/1× phosphate-buffered saline (PBS) solution at 4°C. Serial coronal sections (60 µm) were cut using a vibratome (Leica VT1000S, Leica Biosystems, Nussloch, Germany) and stored in a 0.1 M PBS/0.1% NaN$_3$ solution at 4°C.

Cells expressing eYFP and c-Fos were stained using a modified iDISCO-based protocol (*Pavlova et al., 2018*). Briefly, sections were washed in 1× PBS three times for 10 min each, then dehydrated in 50% MeOH for 2.5 hr. After, sections were washed in 0.2% Tween-20 in PBS (0.2% PBST) three times for 10 min each, and then blocked in a solution of 0.2% PBST, 10% DMSO, and 6% normal donkey serum (NDS) for 2 hr. Sections were washed in 1× PBS, 0.2% Tween-20, and 10 µg heparin (PTwH), and then incubated in a primary antibody solution of rat polyclonal anti-c-Fos (1:5000, SySy, Goettingen, Germany) and chicken polyclonal anti-GFP (10 mg/ml, 1:2000, Abcam, Cambridge, MA) in PTwH/5% DMSO/3% NDS for 48 hr at 4°C. Sections were then washed three times in PTwH for 10 min each, before being incubated in secondary antibody solution consisting of Alexa 647 conjugated Donkey Anti-Rat IgG (1:500, Abcam, Cambridge, MA) and Cy2-conjugated Donkey Anti-Chicken IgG (1:250, Jackson ImmunoResearch, West Grove, PA) in 3% NDS/PTwH overnight. The next day, sections were washed in three increments of 10 min each in PTwH, then washed in three increments of 10 min each in 1× PBS. Sections were mounted on slides and allowed to dry for approximately 20 min before adding mounting medium Fluoromount G (Electron Microscopy Sciences, Hatfield, PA) and a coverslip.

## Confocal microscopy

Coronal brain sections were imaged using a confocal scanning microscope (Leica TCS SP8, Leica Microsystems Inc, Wetzlar, Germany) with two simultaneous PMT detectors. Fluorescence from Cy2 was excited at 488 nm and detected at 500–550 nm, and Alexa Fluor 647 was excited at 634 nm and detected at 650–700 nm. Sections were imaged with a dry Leica 20× objective (NA 0.60, working distance 0.5 mm), with a pixel size of 1.08 × 1.08 µm$^2$, a z step of 3 µm, and z-stack of 9 µm. Fields of view were stitched together to form tiled images by using an automated stage and the tiling function and algorithm of the LAS X software. Raw images were saved as .lif files storing two channels: Cy2 (eYFP, channel 1) and Alexa Fluor 647 (c-Fos, channel 2).

## Automated cell segmentation

Using a custom macro in ImageJ/FIJI, z-stack images of brain sections were converted from .lif to .tif files (*Source code 1*). A previously developed custom 3D segmentation approach was applied to each label due to their differences in morphology (*Leal Santos et al., 2021*). Because c-Fos localizes to the cell body, resulting in punctate filled-spherical staining, an optimized algorithm for this labeling pattern was applied. c-Fos$^+$ cells were identified by first passing the image through a bandpass filter in Fourier space, subtracting the background using a rolling ball algorithm, and identifying the cells using the 3D Local Maxima Fast Filter, 3D Spot Segmentation, and 3D Manager plugins in the 3D ImageJ suite (*Ollion et al., 2013*). Due to the dendritic expression of eYFP after labeling with the ArcCreER$^{T2}$ × eYFP tagging system, an optimized algorithm to minimize segmentation of cellular processes was applied. eYFP$^+$ cells were identified by subtracting the background, blurring the image with a Gaussian kernel, thresholding the image for the 0.5% brightest pixels, and using the thresholded regions as a mask for identifying the cells with the Classic Watershed plugin in the MorphoLibJ suite[9]. The segmentation output was then qualitatively inspected per image, and over-segmentation errors due to structural autofluorescence artifacts (e.g., around ventricles and tissue edges) were manually deleted to maximize accuracy of mapping in the final dataset. Co-labeled cells were identified using the 3D MultiColoc plugin in the 3D ImageJ suite (*Ollion et al., 2013*), which uses the label images created during segmentation of the individual labels to efficiently identify overlapping objects. To maximize precision of counts, co-labeled cells were additionally filtered by the percent volume of overlap between the objects identified in each individual channel, relative to channel 2. A subset of identified co-labeled cells was manually inspected in ImageJ/FIJI to ensure accuracy. Custom ImageJ/FIJI macros were written to apply segmentation and colocalization algorithms in bulk to the images (*Source code 3*, *Source code 4*, *Source code 5*). All macros, along with an example image for trial, are available for download at https://osf.io/ynqp7/, and were generalized to include adjustable parameters to apply to images across different resolutions.

## Manual cell quantification

The number of eYFP$^+$ and c-Fos$^+$ was manually counted by two annotators for the dDG bilaterally across two images per mouse. First, the z-stack image was collapsed into a single maximum projection image using Fiji[6] to facilitate counting. ROIs were drawn around the granule cell layer and saved for comparison with automated cell segmentation results. eYFP$^+$ and c-Fos$^+$ cells were counted within each ROI, with accuracy qualitatively verified by overlaying counts over each section of the original z-stack image. To quantify the number of FN and FP counts, the total number of cells manually quantified per ROI were compared to the collapsed maximum projection image of automated cell counts within the same ROI. Despite the auto-segmentation algorithms being applied in 3D, counting FN and FP cells in 2D for validation, followed by qualitative comparison with the original z-stack, allowed for a more time-efficient workflow for manual comparison. Estimated total TP counts per counter were then calculated by taking the average of two derivations: (Total no. manual counts – Total FN counts) and (Total no. auto-segmented counts – Total FP counts). Individual calculations for precision, recall, and *F*1 were performed for each annotator and then averaged.

## Package development and code availability

The SMARTTR package was developed to bridge the optimized segmentation and colocalization approach in ImageJ/FIJI with the registration and mapping capabilities of the SMART (*Jin et al., 2022*) and WholeBrain (*Fürth et al., 2018*) packages in R. Beyond this, the SMARTTR package offers an extensive suite of functions to analyze and visualize the mapped datasets. The package design intrinsically incorporates data management facilitated by its object-oriented structure. Detailed documentation and in-depth descriptions of package objects and functions are available at the package website (https://mjin1812.github.io/SMARTTR) and access to the source code is available as a GitHub repository (https://github.com/mjin1812/SMARTTR, copy archived at *Jin and Ogundare, 2025*). Additionally, the processed and mapped dataset from the LH experiments is available for download at the package website to follow along an analysis tutorial. SMARTTR was written primarily in base R (v3.6.3) and developed and documented with devtools and roxygen2. Built-in functions for network analyses and visualizations were created using the igraph, tidygraph, ggplot2, and ggraph packages. The SMARTTR website was built using pkgdown.

## Registration and mapping

Image z-stacks were automatically preprocessed using an ImageJ/FIJI macro to combine fluorescence intensities across both channels and z-planes into a flattened maximum project image (*Source code 2*). Images were manually assigned to a best-matching anterior–posterior atlas coordinate based on the reference atlas materials available at openbrainmap.org and https://osf.io/cpt5w. Registration with flexible user correction was then performed in R on these flattened images using the registration() wrapper function in SMARTTR, which interfaces with the WholeBrain (*Fürth et al., 2018*) and SMART (*Jin et al., 2022*) packages.

In brief, WholeBrain allows for registration of serial sections to vectorized coronal reference atlas plates, generated from the 2008 Allen reference atlas. Because of its representation of region boundaries as non-uniform rational B-splines (NURBS), the registration process is scale-invariant in the x- and y-planes and can accommodate a wide range of imaging resolutions. Registration along the z-plane, however, is limited to the spacing between atlas plates (~100 μm), and accuracy depends on qualitative matching to the correct plate. Initial registration occurs by the fitting of 32 correspondence points along the contours of the atlas and tissue sections. These points can be pruned, corrected, or added to through a user-friendly console interface available through the SMART package. The user-corrected correspondence points are then used to calculate the thin-plate splines deformation field, which can interpolate and match any coordinates within the tissue to the atlas. Further details and a tutorial on the user-friendly registration process are available at the SMART package website (https://sgoldenlab.github.io/SMART/). Misaligned or damaged regions per hemisphere were manually annotated by acronym, logged, and excluded from analysis using the exclude_anatomy() function.

Cell counts of eYFP$^+$, c-Fos$^+$, and co-labeled cells were warped to atlas space using the map_cells_to_atlas() function. The area of each region per image was calculated using Gauss's formula and aggregated across all sections in a mouse. Volume of mapped images was calculated by multiplying the aggregated areas by the thickness of each z-stack section. Mapped cell counts were then normalized by region volume to yield cells per mm$^3$. Regions expected not to contain IEG-labeled cells, such as the fiber tracts and ventricles, were excluded from counts in addition to manually curated lists of regions to omit due to tearing, imaging artifacts, etc. Normalized cell counts were stored and saved in the mouse object. Outlier values were detected and removed by finding normalized regional cell counts that were either two standard deviations above or below the group mean.

We improved efficiency by writing a user-friendly notebook for the registration process and using custom scripts to detect misspellings in annotated regions and automate mapping of segmented cell counts onto a standardized atlas space. We have documented all custom scripts in our website tutorial and GitHub repository to facilitate use. Comprehensive statistical mapping results for eYFP$^+$ cell counts, c-Fos$^+$ cell counts, co-labeled/eYFP$^+$, and co-labeled/c-Fos$^+$ reactivation proportions with FDR corrections are listed in *Supplementary file 4*, *Supplementary file 5*, *Supplementary file 6*, and *Supplementary file 7*, respectively.

## Permutation and network analysis

Pearson correlations between regions were calculated using the get_correlations() function, which interfaces with the rcorr() function from the Hmisc package (v4.5.0), and asymptotic p-values for each correlation were determined using a one-sample *t*-test. Significantly correlated regions were visualized in heatmaps using conservative alpha values of p < 0.01 (eYFP$^+$, co-labeled/eYFP$^+$) and p < 0.005 (c-Fos$^+$). Each of these thresholds was chosen to ensure an optimal level of edge sparsity in their respective networks.

Permutation analysis was conducted with the correlation_diff_permutation() function to identify which region correlations differed most between experimental groups. Correlations differences were calculated by subtracting correlations from the CT group from respective correlations in the IS group ($r_{IS} - r_{CT}$). Each correlation difference was used as a test statistic against a null distribution produced by shuffling the labels of CT and IS mice 1000 times, with the correlation difference recomputed for each shuffle. Correlation differences were compared to each individual null distributions to determine the p-value. Significance correlation differences for all analyses were thresholded using an alpha of 0.01 without FDR correction. As our dataset is meant as a demonstrative and exploratory resource, we chose to minimize FN findings at the expense of high tolerance for FPs.

Networks were constructed and visualized using the create_networks() and plot_networks() functions, which interface with the igraph (v1.2.6) and tidygraph (v1.2.0) packages. Because the c-Fos and Arc-driven eYFP labels are both activity dependent, the correlations between regions are akin to functional connections. Thus, significant functional relationships can be visualized by representing regions as nodes and correlations as an edge between two nodes, with the Pearson correlation values determining the weight or thickness of the edge. Since pairwise correlations were calculated for all pairs, the initial functional network is complete, with all nodes fully interconnected. In order to interpret the networks and discover their most salient features, correlations with alpha values below their corresponding heatmap significance thresholds were dropped and additionally filtered to have an $r$ > 0.9 to ensure only the strongest connections were retained. Network topology measures and node centralities were then calculated using the summarize_networks() function. A supplementary community detection analysis was additionally applied to the c-Fos$^+$ and co-labeled/eYFP$^+$ channels using tidygraph functions. Close community modules were identified by calculating the eigenvector of the modularity matrix for the largest positive eigenvalue and then separating vertices into two communities based on the sign of the corresponding element in the eigenvector (*Newman, 2006*).

## Acknowledgements

We thank Dr Sam A Golden for his feedback on an initial iteration of this manuscript. We also thank members of the Denny laboratory for their constructive comments on this project. This work was supported by the National Institute of General Medical Sciences (T32GM007367 to MJ/Columbia MSTP), the National Institute of Child Health and Human Development (R01HD101402 to CAD), the National Institute on Aging (R21AG064774 to CAD; F30AG084312 to MJ), the National Institute of Neurological Disorders and Stroke (R21NS114870 to CAD), and the Marie Sklodowska Curie Action Fellowship (grant no. 101107833 to AF).

## Additional information

### Competing interests

Christine Ann Denny: employee of Research Foundation for Mental Hygiene, Inc (RFMH). The other authors declare that no competing interests exist.

### Funding

| Funder | Grant reference number | Author |
| --- | --- | --- |
| National Institute on Aging | F30AG084312 | Michelle Jin |
| National Institute of General Medical Sciences | T32GM007367 | Michelle Jin |
| Eunice Kennedy Shriver National Institute of Child Health and Human Development | R01HD101402 | Christine Ann Denny |
| National Institute on Aging | R21AG064774 | Christine Ann Denny |
| Marie Sklodowska Curie Action Fellowship | 10.3030/101107833 | Alessandra Franceschini |
| National Institute of Neurological Disorders and Stroke | R21NS114870 | Christine Ann Denny |

The funders had no role in study design, data collection, and interpretation, or the decision to submit the work for publication.

### Author contributions

Michelle Jin, Conceptualization, Data curation, Software, Formal analysis, Supervision, Funding acquisition, Validation, Investigation, Visualization, Methodology, Writing – original draft, Writing – review

and editing; Simon O Ogundare, Data curation, Software, Formal analysis, Investigation, Visualization; Marcos Lanio, Sofia Leal Santos, Conceptualization, Investigation, Methodology; Sophia Sorid, Validation, Investigation; Alicia Ruth Whye, Investigation; Alessandra Franceschini, Methodology; Christine Ann Denny, Conceptualization, Supervision, Funding acquisition, Writing – original draft, Project administration, Writing – review and editing

### Author ORCIDs
Michelle Jin ⓘ https://orcid.org/0000-0002-8696-1958
Simon O Ogundare ⓘ https://orcid.org/0000-0002-6243-2779
Marcos Lanio ⓘ https://orcid.org/0000-0003-3000-1592
Sophia Sorid ⓘ https://orcid.org/0009-0008-7926-1225
Alicia Ruth Whye ⓘ https://orcid.org/0009-0008-0865-8882
Sofia Leal Santos ⓘ https://orcid.org/0000-0001-5552-2517
Alessandra Franceschini ⓘ https://orcid.org/0000-0003-0909-8852
Christine Ann Denny ⓘ https://orcid.org/0000-0002-6926-2020

### Ethics
All experiments were approved by the Institutional Animal Care and Use Committee (IACUC) at the Research Foundation for Mental Hygiene, Inc (RFMH) under protocol No.1522. All key resources are listed in Materials and methods.

Reviewer #1 (Public review): https://doi.org/10.7554/eLife.101327.3.sa1
Reviewer #2 (Public review): https://doi.org/10.7554/eLife.101327.3.sa2
Author response https://doi.org/10.7554/eLife.101327.3.sa3

## Additional files

### Supplementary files
Supplementary file 1. Statistical analysis summary for all behavioral tests.

Supplementary file 2. Comprehensive list of acronyms used.

Supplementary file 3. Statistical analysis summary for targeted brain region analyses.

Supplementary file 4. Statistical analysis summary for eYFP activity across all mapped regions. False discovery rate correction of p-values can be found under the p.adj column.

Supplementary file 5. Statistical analysis summary for c-Fos activity across all mapped regions. False discovery rate correction of p-values can be found under the p.adj column.

Supplementary file 6. Statistical analysis summary for reactivation proportion (co-labeled cells/eYFP+ cells) across all mapped regions. False discovery rate correction of p-values can be found under the p.adj column.

Supplementary file 7. Statistical analysis summary for proportion of reactivation activity (co-labeled cells/c-Fos+ cells) across all mapped regions. False discovery rate correction of p-values can be found under the p.adj column.

Source code 1. Batch creation of TIF files.

Source code 2. Batch creation of maximum projection images.

Source code 3. Batch creation of segmentation of c-Fos+ cells.

Source code 4. Batch creation of segmentation of eYFP+ cells.

Source code 5. Batch detection of co-labeled cells.

MDAR checklist

### Data availability
All example data and scripts are downloadable courtesy of the OSF repository sponsored by the Center for Open Science (COS) at https://osf.io/eacn7/ or https://github.com/mjin1812/SMARTTR (copy archived at *Jin and Ogundare, 2025*). We additionally provide all image processing code as Source codes 1–5. Behavioral data is provided as Figure 2—Source Data 1.

The following dataset was generated:

| Author(s) | Year | Dataset title | Dataset URL | Database and Identifier |
|---|---|---|---|---|
| Jin M | 2025 | SMARTTR Pipeline | https://osf.io/eacn7/ | Open Science Framework, eacn7 |

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
