## [Editor Report · eLife Assessment]

This manuscript describes methods and software, called SMARTR, to map neuronal networks using markers of neuronal activity. They illustrate their approach using tissue from mice that have undergone behavioral tasks. The reviewers considered the study **important** to the field and **compelling** in that the methods and analyses were an advance over current tools.

---

## [Referee Report · Reviewer #1 (Public review)]

Summary:

In this manuscript, Jin et. al., describe SMARTTR, an image analysis strategy optimized for analysis of dual-activity ensemble tagging mouse reporter lines. The pipeline performs cell segmentation, then registers the location of these cells into an anatomical atlas, and finally, calculates the degree of co-expression of the reporters in cells across brain regions. The authors demonstrate the utility of the method by labeling two ensemble populations during two related experiences: inescapable shock and subsequent escapable shock as part of learned helplessness.

Strengths:

- We appreciated that the authors provided all documentation necessary to use their method, and that the scripts in their publicly available repository are well commented. Submission of the package to CRAN will, as the other reviewer pointed out, ensure that the package and its dependencies can be easily installed using few lines of code in the future. Additionally, we particularly appreciate the recently added documentation website and vignettes, which provide guidance on package installation and use cases.

- The manuscript was well-written and very clear, and the methods were generally highly detailed.

- The authors have addressed our previous concerns, and we appreciate their revised manuscript.

---

## [Referee Report · Reviewer #2 (Public review)]

Summary:

This manuscript describes a workflow and software package, SMARTR, for mapping and analyzing neuronal ensembles tagged using activity-dependent methods. They showcase this pipeline by analyzing ensembles tagged during the learned helplessness paradigm. This is an impressive effort, and I commend the authors for developing open-source software to make whole-brain analyses more feasible for the community. After peer-review, the authors addressed reviewer suggestions and concerns regarding the usability and maintainability of the SMARTTR package, ensuring that the package will be published on CRAN, improving documentation, and including unit tests to ensure code stability. Overall, this software package will prove to have a broad impact on the field.

---

## [Author Response]

The following is the authors’ response to the original reviews

**Reviewer #1 (Public review):**
Weaknesses:(1) The heatmaps (for example, Figure 3A, B) are challenging to read and interpret due to their size. Is there a way to alter the visualization to improve interpretability? Perhaps coloring the heatmap by general anatomical region could help? We feel that these heatmaps are critical to the utility of the registration strategy, and hence, clear visualization is necessary.

We thank the reviewers for this point on aesthetic improvement, and we agree that clearer visualization of our correlation heatmaps is important. To address this point, we have incorporated the capability of grouping “child” subregions in anatomical order by their more general “parent” region into the package function, plot_correlation_heatmaps(). Parent regions will be can now be plotted as smaller sub-facets in the heatmaps. We have also rearranged our figures to fit enlarged heatmaps in Figures 3-5, and Supplementary Figure 10 for easier visualization.

(2) Additional context in the Introduction on the use of immediate early genes to label ensembles of neurons that are specifically activated during the various behavioral manipulations would enable the manuscript and methodology to be better appreciated by a broad audience.

We thank the reviewers for this suggestion and have revised the first part of our Introduction to reflect the broader use and appeal of immediate early genes (IEGs) for studying neural changes underlying behavior.

(3) The authors mention that their segmentation strategies are optimized for the particular staining pattern exhibited by each reporter and demonstrate that the manually annotated cell counts match the automated analysis. They mention that alternative strategies are compatible, but don't show this data.

We thank the reviewers for this comment. We also appreciate that integration with alternative strategies is a major point of interest to readers, given that others may be interested in compatibility with our analysis and software package, rather than completely revising their own pre-existing pipelines.

Generally, we have validated the ability to import datasets generated from completely different workflows for segmentation and registration. We have since released documentation on our package website with step-by-step instructions on how to do so (https://mjin1812.github.io/SMARTTR/articles/Part5.ImportingExternalDatasets). We believe this tutorial is a major entry point to taking advantage of our analysis package, without adopting our entire workflow.

This specific point on segmentation refers to the import_segmentation_custom()function in the package. As there is currently not a standard cell segmentation export format adopted by the field, this function still requires some data wrangling into an import format saved as a .txt file. However, we chose not to visually demonstrate this capability in the paper for a few reasons.

i) A figure showing the broad testing of many different segmentation algorithms, (e.g., Cellpose, Vaa3d, Trainable Weka Segmentation) would better demonstrate the efficacy of segmentation of these alternative approaches, which have already been well-documented. However, demonstrating importation compatibility is more of a demonstration of API interface, which is better shown in website documentation and tutorial notebooks.

ii) Additionally, showing importation with one well-established segmentation approach is still a demonstration of a single use case. There would be a major burden-of-proof in establishing importation compatibility with all potential alternative platforms, their specific export formats, which may be slightly different depending on post-processing choices, and the needs of the experimenters (e.g., exporting one versus many channels, having different naming conventions, having different export formats). For example, output from Cellpose can take the form of a NumPy file (_seg.npy file), a .png, or Native ImageJ ROI archive output, and users can have chosen up to four channels. Until the field adopts a standardized file format, one flexible enough to account for all the variables of experimental interest, we currently believe it is more efficient to advise external groups on how to transform their specific data to be compatible with our generic import function.

(4) The authors provided highly detailed information for their segmentation strategy, but the same level of detail was not provided for the registration algorithms. Additional details would help users achieve optimal alignment.

We apologize for this lack of detail. The registration strategy depends upon the WholeBrain (Fürth et al., 2018) package for registration to the Allen Mouse Common Coordinate Framework. While this strategy has been published and documented elsewhere, we have substantially revised our methods section on the registration process to better incorporate details of this approach.

(5) The authors illustrate registration to the Allen atlas. Can they comment on whether the algorithm is compatible with other atlases or with alternative sectioning planes (horizontal/sagittal)?

Since the current registration workflow integrates WholeBrain (Fürth et al., 2018), any limitations of WholeBrain apply to our approach, which means limited support for registering non-coronal sectioning planes and reliance on the Allen Mouse Atlas (Dong, 2008). However, network analysis and plotting functions are currently compatible with the Allen Mouse Brain Atlas and the Kim Unified Mouse Brain Atlas version (2019) (Chon et al., 2019). Therefore, current limitations in registration do not preclude the usefulness of the SMARTTR software in generating valuable insights from network analysis of externally imported datasets.

There are a number of alternative workflows, such as the QUINT workflow (Yates et al., 2019), that support multiple different mouse atlases, and registration of arbitrarily sectioned angles. We have plans to support and a facilitate an entry point for this workflow in a future iteration of SMARTTR, but believe it is of benefit to the wider community to release and support SMARTTR in its current state.

(6) Supplemental Figures S10-13 do not have a legend panel to define the bar graphs.

We apologize for this omission and have fixed our legends in our resubmission. Our supplement figure orders have changed and the corresponding figures are now Supplemental Figures S11-14.

(7) When images in a z-stack were collapsed, was this a max intensity projection or average? Assuming this question is in regards to our manual cell counting validation approach, the zstacks were collapsed as a maximum intensity projection.
**Reviewer #2 (Public review):**
Weaknesses:(1) While I was able to install the SMARTR package, after trying for the better part of one hour, I could not install the "mjin1812/wholebrain" R package as instructed in OSF. I also could not find a function to load an example dataset to easily test SMARTR. So, unfortunately, I was unable to test out any of the packages for myself. Along with the currently broken "tractatus/wholebrain" package, this is a good example of why I would strongly encourage the authors to publish SMARTR on either Bioconductor or CRAN in the future. The high standards set by Bioc/CRAN will ensure that SMARTR is able to be easily installed and used across major operating systems for the long term.

We greatly thank the reviewer for pointing out this weakness; long-term maintenance of this package is certainly a mutual goal. Loading an .RDATA file is accomplished by either doubleclicking directly on the file in a directory window, after specifying this file type should be opened in RStudio or by using the load() function, (e.g., load("directory/example.RData")). We have now explicitly outlined these directions in the online documentation.

Moreover, we have recently submitted our package to CRAN and are currently working on revisions following comments. This has required a package rebranding to “SMARTTR”, as there were naming conflicts with a previously archived repository on CRAN. Currently, SMARTTR is not dependent on the WholeBrain package, which remains optional for the registration portion of our workflow. Ultimately, this independence will allow us to maintain the analysis and visualization portion of the package independently.

In the meantime, we have fully revised our installation instructions (https://mjin1812.github.io/SMARTTR/articles/SMARTTR). SMARTTR is now downloadable from a CRAN-like repository as a bundled .tar.gz file, which should ease the burden of installation significantly. Installation has been verified on a number of different versions of R on different platforms. Again, we hope these changes are sufficient and improve the process of installation.

(2) The package is quite large (several thousand lines include comments and space). While impressive, this does inherently make the package more difficult to maintain - and the authors currently have not included any unit tests. The authors should add unit tests to cover a large percentage of the package to ensure code stability.

We have added unit testing to improve the reliability of our package. Unit tests now cover over 71% of our source code base and are available for evaluation on our github website (https://github.com/mjin1812/SMARTTR). We focused on coverage of the most front-facing functions. We appreciate this feedback, which has ultimately enhanced the longevity of our software.

(3) Why do the authors choose to perform image segmentation outside of the SMARTTR package using ImageJ macros? Leading segmentation algorithms such as CellPose and StarMap have well-documented APIs that would be easy to wrap in R. They would likely be faster as well. As noted in the discussion, making SMARTTR a one-stop shop for multi-ensemble analyses would be more appealing to a user.

We appreciate this feedback. We believe parts of our response to Reviewer 1, Comment 3, are relevant to this point. Interfaces for CellPose and ClusterMap (which processes in situ transcriptomic approaches, like STARmap) are both in python, and currently there are ways to call python from within R (https://rstudio.github.io/reticulate/index.html). We will certainly explore incorporating these APIs from R. However, we would anticipate this capability is more similar to “translation” between programming languages, but would not currently preclude users from the issue of needing some familiarity with the capabilities of these python packages, and thus with python syntax.

(4) Given the small number of observations for correlation analyses (n=6 per group), Pearson correlations would be highly susceptible to outliers. The authors chose to deal with potential outliers by dropping any subject per region that was> 2 SDs from the group mean. Another way to get at this would be using Spearman correlation. How do these analyses change if you use Spearman correlation instead of Pearson? It would be a valuable addition for the author to include Spearman correlations as an option in SMARTTR.

We thank reviewers for this suggestion and we have updated our code base to include the possibility for using Spearman’s correlation coefficient as opposed to Pearson’s correlation coefficient for heatmaps in the get_correlations() function. Users can now use the `method` parameter, set to either “pearson” or “spearman” and results will propagate throughout the rest of the analysis using these results.

Below, in Author response image 1 we show a visual comparison of the correlation heat maps for active eYFP^+^ ensembles in the CT and IS groups using both Pearson and Spearman correlations. We see a strongly qualitative similarity between the heat maps. Of course, since the statistical assumptions underlying the relationship between variables using Pearson correlation (linear) vs Spearman correlation (monotonic) are different, users should take this into account when interpreting results using different approaches.

**Author response image 1. sa3fig1:** Pearson and Spearmen regional correlations of eYFP+ ensembles activity in the CT and IS groups.

(5) I see the authors have incorporated the ability to adjust p-values in many of the analysis functions (and recommend the BH procedure) but did not use adjusted p-values for any of the analyses in the manuscript. Why is this? This is particularly relevant for the differential correlation analyses between groups (Figures 3P and 4P). Based on the un-adjusted pvalues, I assume few if any data points will still be significant after adjusting. While it's logical to highlight the regional correlations that strongly change between groups, the authors should caution which correlations are "significant" without adjusting for multiple comparisons. As this package now makes this analysis easily usable for all researchers, the authors should also provide better explanations for when and why to use adjusted p-values in the online documentation for new users.

We appreciate the feedback note that our dataset is presented as a more demonstrative and exploratory resource for readers and, as such, we accept a high tolerance for false positives, while decreasing risk of missing possible interesting findings. As noted by Reviewer #2, it is still “logical to highlight the regional correlations that strongly change between groups.” We have clarified in our methods that we chose to present uncorrected p-values when speaking of significance.

We have also removed any previous recommendations for preferred methods for multiple comparisons adjustment in our function documentations, as some previous documentation was outdated. Moreover, the standard multiple comparisons adjustment approaches assume complete independence between tests, whereas this assumption is violated in our differential correlational analysis (i.e., a region with one significantly altered connection is more likely than another to have another significantly altered connection).

Ultimately, the decision to correct for multiple comparisons with standard FDR, and choice of significance threshold, should still be informed by standard statistical theory and user-defined tolerance for inclusion of false-positives and missing of false-negatives. This will be influenced by factors, such as the nature and purpose of the study, and quality of the dataset.

(6) The package was developed in R3.6.3. This is several years and one major version behind the current R version (4.4.3). Have the authors tested if this package runs on modern R versions? If not, this could be a significant hurdle for potential users.

We thank reviewers for pointing out concerns regarding versioning. We have since updated our installation approach for SMARTTR, which is compatible with versions of R >= 3.6 and has been tested on Mac ARM-based (Apple silicon) architecture (R v4.4.2), and Windows 10 (R v3.6.3, v4.5.0 [devel]).

The recommendation for users to install R 3.6.3 is primarily for those interested in using our full workflow, which requires installation of the WholeBrain package, which is currently a suggested package. We anticipate updating and supporting the visualization and network analysis capabilities, whilst maintaining previous versioning for the full workflow presented in this paper.

(7) In the methods section: "Networks were constructed using igraph and tidygraph packages." - As this is a core functionality of the package, it would be informative to specify the exact package versions, functions, and parameters for network construction.

We thank reviewers for pointing out the necessity for these details for code reproducibility. We have since clarified our language in the manuscript on the exact functions we use in our analysis and package versions, which we also fully document in our online tutorial. Additionally. We have printed our package development and analysis environment online at https://mjin1812.github.io/SMARTTR/articles/Part7.Development.

(8) On page 11, "Next, we examined the cross-correlations in IEG expression across brain regions, as strong co-activation or opposing activation can signify functional connectivity between two regions" - cross-correlation is a specific analysis in signal processing. To avoid confusion, the authors should simply change this to "correlations".

We thank the reviewer for pointing out this potentially confusing phrasing. We have changed all instances of “cross-correlation” to “correlation”.

(9) Panels Q-V are missing in Figure 5 caption.

We thank the reviewer for pointing out this oversight. We have now fixed this in our revision.

References

Chon, U., Vanselow, D. J., Cheng, K. C., & Kim, Y. (2019). Enhanced and unified anatomical labeling for a common mouse brain atlas. Nature Communications, 10(1), 5067. https://doi.org/10.1038/s41467-019-13057-w

Dong, H. W. (2008). The Allen reference atlas: A digital color brain atlas of the C57Bl/6J male mouse (pp. ix, 366). John Wiley & Sons Inc

Fürth, D., Vaissière, T., Tzortzi, O., Xuan, Y., Märtin, A., Lazaridis, I., Spigolon, G., Fisone, G., Tomer, R., Deisseroth, K., Carlén, M., Miller, C. A., Rumbaugh, G., & Meletis, K. (2018). An interactive framework for whole-brain maps at cellular resolution. Nature Neuroscience, 21(1), 139–149. https://doi.org/10.1038/s41593-017-0027-7

Yates, S. C., Groeneboom, N. E., Coello, C., Lichtenthaler, S. F., Kuhn, P.-H., Demuth, H.-U., Hartlage-Rübsamen, M., Roßner, S., Leergaard, T., Kreshuk, A., Puchades, M. A., & Bjaalie, J. G. (2019). QUINT: Workflow for Quantification and Spatial Analysis of Features in Histological Images From Rodent Brain. Frontiers in Neuroinformatics, 13. https://www.frontiersin.org/articles/10.3389/fninf.2019.00075